# Enhancing Time Series Forecasting through Selective Representation Spaces: A Patch Perspective

**Xingjian Wu, Xiangfei Qiu, Hanyin Cheng, Zhengyu Li,**
**Jilin Hu, Chenjuan Guo, Bin Yang**[*]
East China Normal University
{xjwu,xfqiu,hycheng,lizhengyu}@stu.ecnu.edu.cn,
{jlhu,cjguo,byang}@dase.ecnu.edu.cn

## Abstract

Time Series Forecasting has made significant progress with the help of Patching technique, which partitions time series into multiple patches to effectively retain contextual semantic information into a representation space beneficial for modeling long-term dependencies. However, conventional patching partitions a time series into adjacent patches, which causes a fixed representation space, thus resulting in insufficiently expressful representations. In this paper, we pioneer the exploration of constructing a selective representation space to flexibly include the most informative patches for forecasting. Specifically, we propose the Selective Representation Space (SRS) module, which utilizes the learnable Selective Patching and Dynamic Reassembly techniques to adaptively select and shuffle the patches from the contextual time series, aiming at fully exploiting the information of contextual time series to enhance the forecasting performance of patch-based models. To demonstrate the effectiveness of SRS module, we propose a simple yet effective SRSNet consisting of SRS and an MLP head, which achieves state-of-the-art performance on real-world datasets from multiple domains. Furthermore, as a novel plug-and-play module, SRS can also enhance the performance of existing patch-based models. The resources are available at https://github.com/decisionintelligence/SRSNet.

## 1 Introduction

Time series organize data points chronologically and are either univariate or multivariate depending on the number of variables in each data point. Among the diverse tasks in time series analysis, time series forecasting (TSF) stands out as a critical and widely studied task. It plays a crucial role in various fields such as economics, traffic, energy, and AIOps, providing insights for early warnings and proactive decision-making [Qiu et al., 2024, 2025a, Chen et al., 2024, Wu et al., 2024a,b].

In recent years, significant progress has been made in TSF with the advancement of deep learning technologies. Among these techniques, the method of dividing time series into patches [Cirstea et al., 2022a, Zhang and Yan, 2022, Nie et al., 2023] has gradually gained attention. The significance of dividing time series into patches lies in the fact that a single time step often lacks clear semantic meaning, while the semantic information between adjacent time points tends to be highly similar. Therefore, by performing patch division on the time series, local features and intrinsic patterns can be captured more effectively. In other words, the patching technique introduces an effective way to construct the *representation space* for a contextual time series, which first picks the patches adjacent, then projects them to form the *representations* of the contextal time series. Working upon such *representations* not only enhances the models' ability to understand temporal dependencies but also significantly reduces computational complexity, thereby improving overall prediction efficiency.

---

[*]Corresponding Author

39th Conference on Neural Information Processing Systems (NeurIPS 2025).

However, the commonly-used adjacent patching technique, i.e., with fixed stride, divides patches on different contextual time series with the same positions, which results in a fixed *representation space*. Though multi-scale modeling [Chen et al., 2024, Tang and Zhang, 2025a] may create several representation spaces with different patch sizes, the fixed strides also limit the potential. Because fixed representation spaces assume that all information useful for forecasting is evenly distributed in the contextual time series. As shown in the upper part of Figure 1, the assumption is broken due to the phenomenons of changeable periods, shifting, and anomalies. In such cases, the conventional adjacent patching may break the semantics of periods or include anomalies and the shifting processes, which retains information that may be harmful for forecasting into the *representation space*. Since the purpose of patching technique is to construct appropriate representations for contextual time series, allowing models to adaptively include useful information for forecasting is a more reasonable solution. Therefore, this calls for a *selective representation space* to mitigate the above phenomenons, which relies on *a more flexible and adaptive patching paradigm*.

Inspired by the aboved motivations, we propose **SRS**, a plug-and-play module to efficiently construct the **S**elective **R**epresentation **S**pace for contextual time series. Technically, we propose the *Selective Patching* to select the patches from

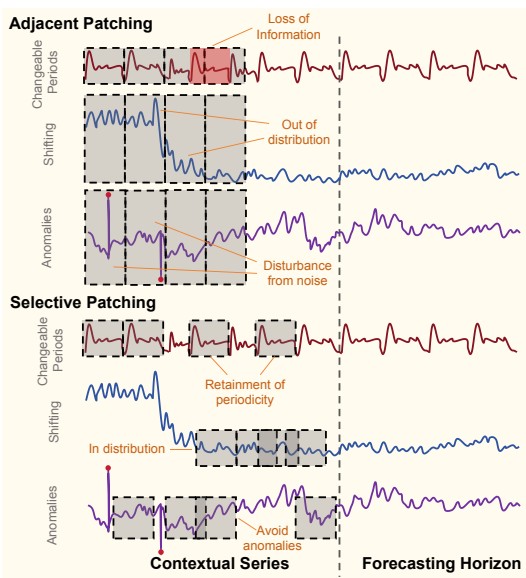

Figure 1: Adjacent vs. Selective Patching (both using 4 patches). Adjacent patching partitions time series into adjacent patches, lacking flexibility. Selective patching automatically select most relevant sub-series as patches. The upper part shows examples that conventional adjacent patching may include harmful information, thus hindering the forecasting performance. The lower part shows the selected patches that are more relevant for making the corresponding forecasting, demonstrating the flexibility that selective patches offer.

the contextual time series, and utilize the *Dynamic Reassembly* to determine the sequence of patches. Both the *Selective Patching* and *Dynamic Reassembly* are designed through a gradient-friendly paradigm and optimized by the objective of forecasting tasks to pursue better prediction accuracy, thus fully exploiting the information in the contextual time series in an adaptive way. We also propose a simple yet effective **SRSNet** composed of SRS and an MLP head. Through comprehensive experiments on datasets from multiple domains, SRSNet achieves state-of-the-art performance to demonstrate the SRS's strong capability of information integration. The contributions are summarized as follows:

- We propose a modular SRS, which efficiently and adaptively constructs the selective representation space to fully exploit the information in the contextual time series, and is able to improve the performance of patch-based models in a simple plug-and-play paradigm.

- Technically, we devise the Selective Patching and Dynamic Reassembly techniques, which are able to constitute a selective representation space at the patch perspective. And they are easily to be optimized through gradient-based strategies.

- Applying SRS with an MLP forms a simple-yet-effective method, called SRSNet. SRSNet achieves state-of-the-art performance across multiple real-world datasets, demonstrating the effectiveness of SRS module.

## 2 Related works

### 2.1 Development of Time Series Forecasting

Time series forecasting (TSF) predicts future observations based on historical observations. Early proposals primarily employed statistical learning methods. ARIMA [Box and Pierce, 1970], ETS [Hyndman et al., 2008], and VAR [Godahewa et al., 2021] are classical and widely utilized methods. With

the rapid progress in machine learning technologies, new methods for TSF leveraging machine learning have been developed [Fischer et al., 2020]. Notably, XGBoost [Chen and Guestrin, 2016], Random Forests [Breiman, 2001] and LightGBM [Ke et al., 2017] have been applied extensively to better accommodate nonlinear relationships and complex patterns. However, these methods still require manual feature engineering and model design. Taking advantage of the representation learning capabilities offered by deep neural networks (DNNs) on rich data, numerous deep learning-based methods have been proposed. TimesNet [Wu et al., 2023a] and SegRNN [Lin et al., 2023] treat time series data as sequences of vectors and utilize CNNs or RNNs to capture temporal dependencies. Transformer architectures, including Informer [Zhou et al., 2021], FEDformer [Zhou et al., 2022], Autoformer [Wu et al., 2021], Triformer [Cirstea et al., 2022b], and PatchTST [Nie et al., 2023] can capture complex relationships between time points more accurately, significantly improving forecasting performance. MLP-based methods, including SparseTSF [Lin et al., 2024a], CycleNet [Lin et al., 2024b], DUET [Qiu et al., 2025a], NLinear [Zeng et al., 2023], and DLinear [Zeng et al., 2023], utilize relatively straightforward architectures with a reduced number of parameters. Nevertheless, they have demonstrated highly competitive performance in forecasting accuracy.

## 2.2 Progress in Patch-based Time Series Forecastng Methods

Patching is a technique originally inspired by the Vision Transformer [Dosovitskiy et al., 2021] and was first introduced in the context of time series forecasting by Triformer [Cirstea et al., 2022a] and PatchTST [Nie et al., 2023]. In Triformer and PatchTST, time series are segmented into subseries-level patches, which are then treated as input tokens to the Transformer, allowing for modeling temporal dependencies at the patch level. Crossformer [Zhang and Yan, 2022] further extends this idea by segmenting each time series into patches and employing self-attention mechanisms to model dependencies across both variables and time dimensions. xPatch [Stitsyuk and Choi, 2025a] adopts the same patch segmentation strategy as PatchTST but introduces a dual-flow architecture consisting of an MLP-based linear stream and a CNN-based nonlinear stream. This design explores the advantages of combining patching with channel-independence techniques within a non-transformer framework. Pathformer [Chen et al., 2024] and PatchMLP [Tang and Zhang, 2025b] delve deeper into the effectiveness of patches in time series forecasting and adopt Multi-Scale Patch Embedding. This method effectively captures multiscale relationships within input sequences, providing a more enriched representation for downstream forecasting tasks. Nevertheless, current methods overlook issues related to fixed patch strides, which results in fixed representation spaces and may lead to information loss or include unhelpful information. In this paper, we introduce the Selective Representation Space (SRS) module for time series data, which adaptively selects patches and determines their sequences to fully exploit the information in the contextual time series.

## 3 Methodology

Given a contextual time series $X \in \mathbb{R}^{N \times T}$ with $N$ channels and $T$ observations, the objective of time series forecasting is to predict the target future horizon $Y \in \mathbb{R}^{N \times L}$ with $L$ steps. The *Representation Space* is constructed through the patching & embedding operations upon the contextual time series, which first choose the representative subsequences, i.e., patches, and then obtain their embeddings. Recent methods adopt adjacent patching technique [Nie et al., 2023, Zhang and Yan, 2022, Stitsyuk and Choi, 2025b, Tang and Zhang, 2025a, Chen et al., 2024, Sun et al., 2025a, Niu et al., 2025], which leads to fixed *representation space*s for different contextual time series by retrieving patches from the same indices. Our proposed Selective Representation Space (SRS) Module (Figure 2), as a plug-and-play technique, aims at making full use of the contextual time series by constructing flexible *Selective Representation Space*s to enhance the performance of patch-based models.

### 3.1 Structure Overview

Note that the time series data is always first processed through the Instance Normalization to mitigate the statistical differences between training and testing parts, which reduces the difficulty of model generalization. We then describe the overall pipeline of the Selective Representation Space (SRS) module in Figure 2 by considering the data flow of an input contextual time series $X \in \mathbb{R}^{N \times T}$.

We first reintroduce the conventional patching [Cirstea et al., 2022a, Nie et al., 2023], i.e., the adjacent patching. Given the patch size $p$ and stride length $s$, the contextual time series $X \in \mathbb{R}^{N \times T}$

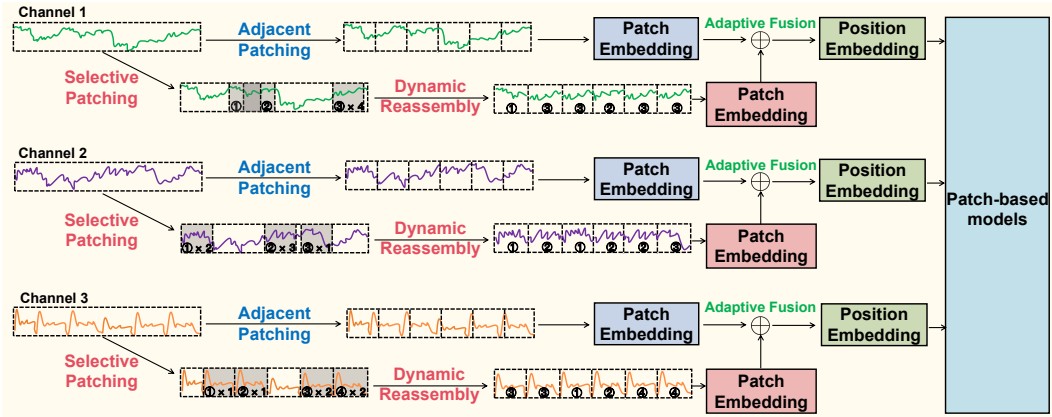

Figure 2: The overall pipeline of the SRS module. The multivariate time series is processed with Channel Independent strategy, the Selective Patching first adaptively chooses proper patches from all potential candidate patches. Then the Dynamic Reassembly dertermines the order of the selected patches. Both the Selective Patching and Dynamic Reassembly are gradient-based and learnable. Finally, the Adaptive Fusion integrates the embeddings from Conventional Patching and Dynamic Reassembly, adds the position embeddings to construct the final representations. The subsequent backbones can be used directly without changes, so that the SRS module is a modular plugin.

is first padded as $X' \in \mathbb{R}^{N \times [p+(n-1)\cdot s]}$, and then reorganized as $n = \lceil (T-p)/s \rceil + 1$ patches $\mathcal{P} \in \mathbb{R}^{N \times n \times p}$. The Conventional Patching stably fetches the patches of fixed indices in different contextual time series, while SRS module aims at automatically selecting patches for each contextual time series.

Considering all potential patch candidates in the padded contextual time series $X' \in \mathbb{R}^{N \times [p+(n-1)\cdot s]}$, there totally exists $K = (n-1)\cdot s + 1$ patches. Instead of selecting $n$ fixed ones with equal space, our proposed *Selective Patching* technique is able to adaptively choose $n$ patches to spontaneously utilize the information from the contextual time series, which is continously optimized during backpropagation. The *Selective Patching* also allows repeated selection, which means that beneficial patches can be selected more than one time. This helps the SRS module "imagine" a more suitable and useful representation space for forecasting. Statistically, there exists $C_{K+n-1}^{n}$ potential options to construct a representation space. We introduce the details of the *Selective Patching* in Section 3.2.

Since the essence of SRS module is to reorganize the contextual series which treats the patches as the basic units. We study to what extent the order of selected patches matters, because most methods adopt permutation-variant components which are sensitive to the order. To this end, the *Dynamic Reassembly* tenchnique is designed to adaptively determine the order of the $n$ selected patches of each contextual time series, which statistically provides $n!$ potential options. The *Dynamic Reassembly* is also learnable, and the details are introduced in Section 3.3.

In summary, the *Selective Patching* and *Dynamic Reassembly* jointly construct a search space with the size of $C_{K+n-1}^{n} \cdot n!$, and they are optimized efficiently through a gradient-based strategy to explore such huge search space. Finally, we conduct an *Adaptive Fusion* during the embedding phase to adaptively fuse the embeddings from the Conventional Patching and our method, making them complement each other. We introduce it in Section 3.4. Working as a plug-and-play technique, SRS Module can be easily integrated into patch-based backbones.

## 3.2 Selective Patching

The objective of *Selective Patching* is to adaptively choose $n$ patches with size $p$ from the padded contextual time series $X' \in \mathbb{R}^{N \times [p+(n-1)\cdot s]}$, to align with the Conventional Patching. As shown in Figure 3 left, we scan the contextual time series with stride equals 1, where all potential patches are represented as $\mathcal{P}' \in \mathbb{R}^{N \times K \times p}, K = (n-1)\cdot s + 1$.

To adaptively choose $n$ patches from the total $K$ ones, we devise an MLP-based $Scorer^s$ to score each patch, and then select the one with the highest score based on the ranking. The sample-wised

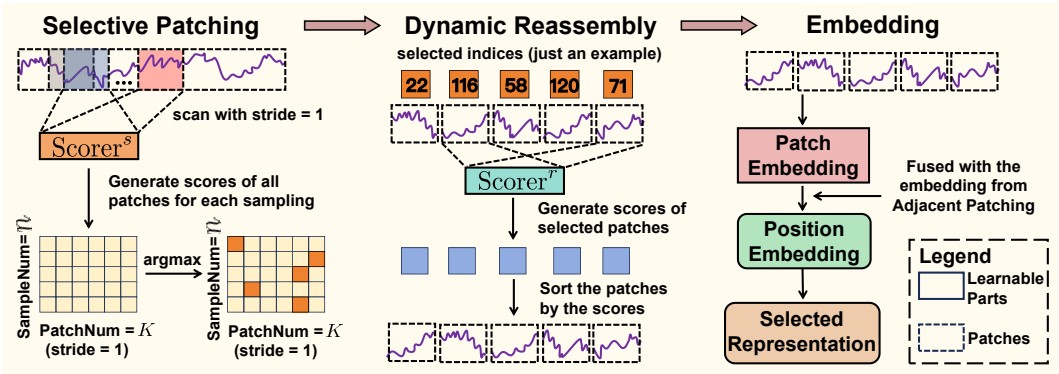

Figure 3: The detailed architecture of the SRS module. The Selective Patching allows sampling with replacement. It scans all the potential patches with stride equals 1, generates $n$ scores for each, then retrieves the patches with max scores in each sampling. Then the Dynamic Reassembly generates scores for selected patches, and sorts them based on the scores to determine the sequence. In the Embedding phase, both the embeddings from the Dynamic Reassembly and Conventional Patching are adaptively fused to form the representations.

process is inspired by Selective State Space Models such as Mamba [Dao and Gu, 2024], which maintains real-time scores based on the data characteristics. Furthermore, to support selection with replacement, we generate $n$ scores for each patch, denoting the scores of $n$ times of sampling:

$$\text{Scorer}^s := \mathbb{R}^{N \times K \times p} \to \mathbb{R}^{N \times K \times n}, \tag{1}$$

$$\mathcal{S}^s = \text{Scorer}^s(\mathcal{P}'), \mathcal{I}^s = \text{Argmax}(\mathcal{S}^s), \tag{2}$$

where $\mathcal{S}^s \in \mathbb{R}^{N \times K \times n}$ denotes the scores of $n$ times of sampling, and $\mathcal{I}^s \in \mathbb{R}^{N \times 1 \times n}$ dentoes the indices of patches with maximal scores. However, the Argmax operation interrupts the gradient propagation, and existing soft sorting methods [Cuturi et al., 2019, Blondel et al., 2020] keep gradient propagation but introduce noise and inaccuracies, making the sorting non-intuitive. To mitigate this, we devise a method to achieve differentiable sorting and keep accurate as the traditional methods. Note that gradients are attached to $\mathcal{S}^s$, we intuitively reuse it with a simple yet effective method:

$$\mathcal{S}^s_{max} = \mathcal{S}^s[\mathcal{I}^s], \mathcal{S}^s_{inv} = \text{detach}(1/\mathcal{S}^s_{max}), \tag{3}$$

$$\mathcal{P}^s_{max} = \mathcal{P}'[\mathcal{I}^s], E^s = \mathcal{S}^s_{max} \odot \mathcal{S}^s_{inv}, \tag{4}$$

$$\tilde{\mathcal{P}}^s_{\max} = \mathcal{P}^s_{max} \odot E^s, \tag{5}$$

where $[\cdot]$ means retrieving values based on the indices, $\mathcal{S}^s_{max}, \mathcal{S}^s_{inv} \in \mathbb{R}^{N \times n}$ denote the maximum scores and their reciprocal. We take the Hadamard product of the selected patches $\mathcal{P}^s_{max} \in \mathbb{R}^{N \times n \times p}$ with $E^s \in \mathbb{R}^{N \times n}$ to attach the gradients to $\tilde{\mathcal{P}}^s_{\max} \in \mathbb{R}^{N \times n \times p}$ through tensor broadcast along the dimension of patch size. Obviously, $E^s$ is an all-one matrix with gradients, because we detach $\mathcal{S}^s_{inv}$ from the calculation graph to make a hard connection to create the paths of gradients.

### 3.3 Dynamic Reassembly

After obtaining the selected patches $\tilde{\mathcal{P}}^s_{max} \in \mathbb{R}^{N \times n \times p}$, the order of them is further considered to enhance the construction of the representation space. As shown in Figure 3 middle, the *Dynamic Reassembly* is devised to adaptively learn an order to reorganize the selected patches.

Following the design of *Selective Patching*, we also utilize an MLP-based Scorer$^r$ to score the selected patches $\tilde{\mathcal{P}}^s_{max} \in \mathbb{R}^{N \times n \times p}$. Different from *Selective Patching*, the *Dynamic Reassembly* processes the permutation problem so that the Scorer$^r$ only generates one score for each selected patch:

$$\text{Scorer}^r := \mathbb{R}^{N \times n \times p} \to \mathbb{R}^{N \times n}, \tag{6}$$

$$\mathcal{S}^r = \text{Scorer}^r(\tilde{\mathcal{P}}^s_{max}), \mathcal{I}^r = \text{Argsort}(\mathcal{S}^r), \tag{7}$$

where $\mathcal{S}^r \in \mathbb{R}^{N \times n}$ denotes the scores of selected patches, $\mathcal{I}^r \in \mathbb{R}^{N \times n}$ denotes the indices sorted by the scores $\mathcal{S}^r$. We then sort the patches according the indices $\mathcal{I}^r$ and use the same technique in

*Selective Patching* to keep gradient propagation:

$$\mathcal{S}^r_{sort} = \mathcal{S}^r[\mathcal{I}^r], \mathcal{S}^r_{inv} = \text{detach}(1/\mathcal{S}^r_{sort}), \tag{8}$$

$$\mathcal{P}^r_{sort} = \tilde{\mathcal{P}}^s_{max}[\mathcal{I}^r], E^r = \mathcal{S}^r_{sort} \odot \mathcal{S}^r_{inv}, \tag{9}$$

$$\tilde{\mathcal{P}} = \mathcal{P}^r_{sort} \odot E^r, \tag{10}$$

where $\mathcal{S}^r_{sort}, \mathcal{S}^r_{inv} \in \mathbb{R}^{N \times n}$ denote the sorted scores and their reciprocal. The $\mathcal{P}^r_{sort} \in \mathbb{R}^{N \times n \times p}$ denotes the sorted patches, and $\tilde{\mathcal{P}} \in \mathbb{R}^{N \times n \times p}$ denotes the sorted patches attached with gradients. $E^r \in \mathbb{R}^{N \times n}$ is obtained by the same way as $E^s$.

## 3.4 Embedding

Through *Selective Patching* and *Dynamic Reassembly*, $\tilde{\mathcal{P}} \in \mathbb{R}^{N \times n \times p}$ are the finally determined patches to represent the contextual time series $X \in \mathbb{R}^{N \times T}$, which also determines the structure of the representation space. We then manage to obtain the embeddings of these patches to feed the patch-based backbones.

To fully utilize the information from both the constructed patches $\tilde{\mathcal{P}} \in \mathbb{R}^{N \times n \times p}$ and the conventional adjacent patches $\mathcal{P} \in \mathbb{R}^{N \times n \times p}$, we design the *Adaptive Fusion* to fuse their patch embeddings. Technically, we construct a convex combination between their patch embeddings:

$$\text{PatchEmbedding}^1, \text{PatchEmbedding}^2 := \mathbb{R}^{N \times n \times p} \to \mathbb{R}^{N \times n \times d}, \tag{11}$$

$$\mathcal{E}^c = \text{PatchEmbedding}^1(\mathcal{P}), \mathcal{E}^s = \text{PatchEmbedding}^2(\tilde{\mathcal{P}}), \tag{12}$$

$$\tilde{\mathcal{E}} = \alpha \odot \mathcal{E}^c + (1 - \alpha) \odot \mathcal{E}^s, \tag{13}$$

where PatchEmbedding denotes a linear projection and is commonly used in time series analysis to encode the patches as embeddings. For embeddings from conventional patching $\mathcal{E}^c \in \mathbb{R}^{N \times n \times d}$ and our proposed method $\mathcal{E}^s \in \mathbb{R}^{N \times n \times d}$, they are balanced with $\alpha \in [0, 1]^{n \times d}$ (broadcast) to work in a complementary manner. Note that $\alpha \in [0, 1]^{n \times d}$ is the most suitable case beacuse the channel dimension $N$ may be vague in some special circumstances such as pretraining. Finally, we add the PositionEmbedding such as Sinusoidal Embedding [Vaswani et al., 2017] to supplement location information.

$$\mathcal{E} = \tilde{\mathcal{E}} + \text{PositionEmbedding}(\tilde{\mathcal{P}}), \tag{14}$$

where the $\mathcal{E} \in \mathbb{R}^{N \times n \times d}$ are the embeddings generated by our proposed SRS Module. Subsequently, they are fed into the patch-based backbones [Tang and Zhang, 2025a, Nie et al., 2023, Zhang and Yan, 2022, Stitsyuk and Choi, 2025b] to extract the interrelationship. These backbones often consist of an Encoder to further extract the representaions of these embeddings, and a Decoder (linear heads or autoregressive heads) to make forecasts based on the representations. SRS Module directly affects the representation space of Encoders through an adaptive way, so that it is named as "Selective Representation Space".

## 4 SRSNet

To demonstrate the strong ability of the SRS module to fully utilize contextual time series at a Patch Perspective, we design a simple yet effective model named **SRSNet**. It consists of SRS module and a Linear/MLP Head ($\leq 2$ layers), which can theoretically fit any continuous function according to the universal approximation theorem [Lu et al., 2021]. SRSNet is trained through Mean Squared Error (MSE) for forecasting tasks:

$$\text{Flatten} := \mathbb{R}^{N \times n \times d} \to \mathbb{R}^{N \times (n \times d)}, \text{MLP} := \mathbb{R}^{N \times (n \times d)} \to \mathbb{R}^{N \times L}, \tag{15}$$

$$\hat{Y} = \text{MLP}(\text{Flatten}(\mathcal{E})), \text{Loss} = ||Y - \hat{Y}||_2^2 \tag{16}$$

As shown in Section 5, the strong performance of SRSNet demonstrates that SRS module makes full use of the contextual time series, and performs well without a complex backbone.

# 5 Experiments

## 5.1 Experimental Details

**Datasets** To conduct comprehensive and fair comparisons for different models, we conduct experiments on eight well-known forecasting benchmarks as the target datasets, including ETT (4 subsets), Weather, Electricity, Solar, and Traffic, which cover multiple domains–see Table 1.

Table 1: Statistics of datasets.

| Dataset | Domain | Frequency | Lengths | Dim | Split | Description |
|---|---|---|---|---|---|---|
| ETTh1 | Electricity | 1 hour | 14,400 | 7 | 6:2:2 | Power transformer 1, comprising seven indicators such as oil temperature and useful load |
| ETTh2 | Electricity | 1 hour | 14,400 | 7 | 6:2:2 | Power transformer 2, comprising seven indicators such as oil temperature and useful load |
| ETTm1 | Electricity | 15 mins | 57,600 | 7 | 6:2:2 | Power transformer 1, comprising seven indicators such as oil temperature and useful load |
| ETTm2 | Electricity | 15 mins | 57,600 | 7 | 6:2:2 | Power transformer 2, comprising seven indicators such as oil temperature and useful load |
| Weather | Environment | 10 mins | 52,696 | 21 | 7:1:2 | Recorded every for the whole year 2020, which contains 21 meteorological indicators |
| Electricity | Electricity | 1 hour | 26,304 | 321 | 7:1:2 | Electricity records the electricity consumption in kWh every 1 hour from 2012 to 2014 |
| Solar | Energy | 10 mins | 52,560 | 137 | 6:2:2 | Solar production records collected from 137 PV plants in Alabama |
| Traffic | Traffic | 1 hour | 17,544 | 862 | 7:1:2 | Road occupancy rates measured by 862 sensors on San Francisco Bay area freeways |

**Baselines** We compare SRSNet against state-of-the-art models in recent years, including TimeKAN [Huang et al., 2025a], Amplifier [Fei et al., 2025], iTransformer [Liu et al., 2024], TimeMixer [Wang et al., 2024], PatchTST [Nie et al., 2023], Crossformer [Zhang and Yan, 2022], TimesNet [Wu et al., 2023a], DLinear [Zeng et al., 2023], Non-stationary Transformer (Stationary) [Liu et al., 2022], and Fedformer [Zhou et al., 2022].

**Implementation Details** To keep consistent with previous works, we adopt Mean Squared Error (mse) and Mean Absolute Error (mae) as evaluation metrics. We consider four forecasting horizon $F$: {96, 192, 336, 720} for all datasets. Since the size of the look-back window can affect the performance of different models, we choose the look-back window size in {96, 336, 512} for all datasets and report each method's best results for fair comparisons.

## 5.2 Main Results

Comprehensive forecasting results are listed in Table 2 with the best in red bold and the second blue underlined. We have the following observations: 1) Compared with forecasters of different structures (Transformer, CNN, KAN, MLP, Linear), SRSNet achieves an excellent predictive performance. It achieves most first rankings in average results and results of different forecastings horizons (Table 8 in Appendix A). 2) Compared to other methods with fixed representation space constructed through conventional adjacent patching (such as PatchTST and Crossformer), SRSNet introduces an adaptive patching mechanism to construct selective representation spaces, demonstrating superior performance across a wider range of datasets. To demonstrate the selective representation spaces that are more helpful for forecasting, we offer intuitive visualization results in Figure 5-7 in Appendix 5, which demonstrates the capability of SRS in selecting most relevant patches for subsequent forecasting. In summary, with the innovative design of SRS Module, even the simple SRSNet (SRS + MLP) can flexibly handle datasets with diverse characteristics, thus standing out in various scenarios.

Table 2: Multivariate forecasting average results with forecasting horizons $F \in \{96, 192, 336, 720\}$ for the datasets. **Red**: the best, Blue: the 2nd best. Full results are available in Table 8 of Appendix A.

| Datasets | ETTh1 | | ETTh2 | | ETTm1 | | ETTm2 | | Weather | | Electricity | | Solar | | Traffic | |
|---|---|---|---|---|---|---|---|---|---|---|---|---|---|---|---|---|
| Metrics | mse | mae | mse | mae | mse | mae | mse | mae | mse | mae | mse | mae | mse | mae | mse | mae |
| FEDformer [2022] | 0.433 | 0.454 | 0.406 | 0.438 | 0.567 | 0.519 | 0.335 | 0.380 | 0.312 | 0.356 | 0.219 | 0.330 | 0.641 | 0.628 | 0.620 | 0.382 |
| Stationary [2022] | 0.667 | 0.568 | 0.377 | 0.419 | 0.531 | 0.472 | 0.384 | 0.390 | 0.287 | 0.310 | 0.194 | 0.295 | 0.390 | 0.387 | 0.622 | 0.340 |
| DLinear [2023] | 0.430 | 0.443 | 0.470 | 0.468 | 0.356 | 0.378 | 0.259 | 0.324 | 0.242 | 0.295 | 0.167 | 0.264 | 0.224 | 0.286 | 0.418 | 0.287 |
| TimesNet [2023] | 0.468 | 0.459 | 0.390 | 0.417 | 0.408 | 0.415 | 0.292 | 0.331 | 0.255 | 0.282 | 0.190 | 0.284 | 0.211 | 0.281 | 0.617 | 0.327 |
| Crossformer [2023] | 0.439 | 0.461 | 0.894 | 0.680 | 0.464 | 0.456 | 0.501 | 0.505 | 0.232 | 0.294 | 0.171 | 0.263 | 0.205 | **0.232** | 0.522 | 0.282 |
| PatchTST [2023] | 0.419 | 0.436 | 0.351 | 0.395 | 0.349 | 0.381 | 0.256 | 0.314 | 0.224 | **0.262** | 0.171 | 0.270 | 0.200 | 0.284 | 0.397 | 0.275 |
| TimeMixer [2024] | 0.427 | 0.441 | 0.347 | 0.394 | 0.356 | 0.380 | 0.257 | 0.318 | 0.225 | 0.263 | 0.185 | 0.284 | 0.203 | 0.261 | 0.410 | 0.279 |
| iTransformer [2024] | 0.440 | 0.445 | 0.359 | 0.396 | 0.347 | 0.378 | 0.258 | 0.318 | 0.232 | 0.270 | 0.163 | 0.258 | 0.202 | 0.260 | 0.397 | 0.281 |
| Amplifier [2025] | 0.421 | 0.433 | 0.356 | 0.402 | 0.353 | 0.379 | 0.256 | 0.318 | **0.223** | 0.264 | 0.163 | 0.256 | 0.202 | 0.256 | 0.417 | 0.290 |
| TimeKAN [2025] | 0.409 | 0.427 | 0.350 | 0.397 | **0.344** | 0.380 | 0.260 | 0.318 | 0.226 | 0.268 | 0.164 | 0.258 | 0.198 | 0.263 | 0.420 | 0.286 |
| SRSNet | **0.404** | **0.424** | **0.334** | **0.385** | 0.351 | **0.378** | **0.252** | **0.314** | 0.226 | 0.266 | **0.161** | **0.254** | **0.183** | 0.239 | **0.392** | **0.270** |

## 5.3 Ablation Study and Analysis

To investigate the effectiveness of SRS, we conduct comprehensive experiments on four datasets from multiple domains with different lengths and numbers of variables: ETTh1, ETTm2, Solar, and Traffic. The experiments include two parts, one is to analyze the effectiveness of SRS to enhance the prediction accuracy of patch-based models, and the other is to analyze the influences of different components in SRS, i.e., Selective Patching, Dynamic Reassembly, and Adaptive Fusion.

**Ablation study of SRS** To demonstrate the SRS's capablities of working as a modular plugin to enhance the prediction accuracy of patch-based models, we combine SRS with naive MLP, i.e., SRSNet, classic models Crossformer [Zhang and Yan, 2022] and PatchTST [Nie et al., 2023], and recent state-of-the-art models xPatch [Stitsyuk and Choi, 2025b] and PatchMLP [Tang and Zhang, 2025a]. The average results of different horizons are shown in Table 3.

First, when combining SRS with a naive MLP, a significant improvement of prediction accuracy (approximately 5% to 15%) is observed. This demonstrates the impressive performance of SRSNet comes from the SRS module, which helps fully exploit the information of contextual time series by adaptively constructing a selective representation space with Selective Patching and Dynamic Reassembly. It also indicates that consitituting a superio representation space is somewhat more efficient and easier than designing complex models to refine representations in a mediocre representation space. Because the former aims at lightening the burden on models by providing more effective representations, thus especially effective for simple MLPs due to the universal approximation theorem [Lu et al., 2021], while the latter hopes model itself to extract the intricate inductive bias in data, thus unstable to different scenarios, e.g., Crossformer performs well on Solar while poorly on ETTm2. This may be the main reason why SRSNet gains the most improvements than other models.

Table 3: Ablation study of SRS. Five models are considered to show the effectiveness of SRS to work as a plugin. **Black**: the improvement. Full results are available in Table 11–12 of Appendix A.

| Datasets | ETTh1 | | ETTm2 | | Solar | | Traffic | |
|---|---|---|---|---|---|---|---|---|
| Metrics | MSE | MAE | MSE | MAE | MSE | MAE | MSE | MAE |
| MLP | 0.430 | 0.451 | 0.273 | 0.336 | 0.219 | 0.277 | 0.413 | 0.284 |
| + SRS | 0.404 | 0.424 | 0.252 | 0.315 | 0.183 | 0.239 | 0.392 | 0.270 |
| Improve | **6.05%** | **6.06%** | **7.70%** | **6.31%** | **16.08%** | **13.47%** | **5.26%** | **4.88%** |
| PatchTST | 0.419 | 0.436 | 0.256 | 0.314 | 0.200 | 0.284 | 0.397 | 0.275 |
| + SRS | 0.404 | 0.426 | 0.249 | 0.307 | 0.182 | 0.251 | 0.386 | 0.266 |
| Improve | **3.48%** | **2.20%** | **3.00%** | **2.33%** | **8.87%** | **11.13%** | **2.74%** | **3.34%** |
| Crossformer | 0.439 | 0.461 | 0.501 | 0.505 | 0.205 | 0.232 | 0.522 | 0.282 |
| + SRS | 0.432 | 0.455 | 0.454 | 0.462 | 0.193 | 0.225 | 0.512 | 0.274 |
| Improve | **1.55%** | **1.45%** | **9.83%** | **8.22%** | **5.66%** | **2.92%** | **1.85%** | **2.84%** |
| PatchMLP | 0.435 | 0.443 | 0.261 | 0.322 | 0.193 | 0.250 | 0.413 | 0.287 |
| + SRS | 0.422 | 0.436 | 0.253 | 0.315 | 0.179 | 0.242 | 0.402 | 0.277 |
| Improve | **2.99%** | **1.55%** | **2.94%** | **2.31%** | **7.00%** | **3.11%** | **2.57%** | **3.41%** |
| xPatch | 0.416 | 0.429 | 0.253 | 0.308 | 0.186 | 0.210 | 0.398 | 0.248 |
| + SRS | 0.406 | 0.422 | 0.244 | 0.303 | 0.179 | 0.204 | 0.389 | 0.240 |
| Improve | **2.38%** | **1.66%** | **3.59%** | **1.86%** | **4.01%** | **2.61%** | **2.17%** | **3.38%** |

Second, we further demonstrate that SRS is also useful for other patch-based models by providing a better representation space. It is observed that incoerporating SRS still improves the prediction accuarcy of existing patch-based models, with an average approximately 5% improvement on PatchTST, Crossformer, xPatch, and PatchMLP. And it is also orthogonal to multi-scale techniques. For PatchMLP which utilizes different sizes of patches to capture the multi-scale structural information, SRS can be incorporated in the patch embedding part of each scale to pursue a more flexible construction of representation spaces.

**Ablation study of key components in SRS** As SRS consists of Selective Patching, Dynamic Reassembly, and Adaptive Fusion, it is important to analyze the influences of these components. As shown in Table 4, we conduct experiments on four variants, eliminating the three key components respectively (w/o Selective Patching, w/o Dynamic Reassembly, and w/o Adaptive Fusion) to verify their effectiveness, analyzing their collaboration capability by comparing with the naive MLP (w/o SRS).

Experimental results show that the Selective Patching consistently has the greatest impact. Theoretically, Selective Patching determines which patches to be chosen to represent a contextual time series, mainly affecting the construction of the representation spaces. The Dynamic Reassembly and Adaptive Fusion are also effective. The Dynamic Reassembly supports SRS to adaptively reassemble the selected patches to provide

Table 4: Ablation Study of key components in SRS. Four key components in SRS are verified as to whether they are effective. **Red**: the best. Full results are available in Table 9–10 of Appendix A.

| Datasets | ETTh1 | | ETTm2 | | Solar | | Traffic | |
|---|---|---|---|---|---|---|---|---|
| Metrics | MSE | MAE | MSE | MAE | MSE | MAE | MSE | MAE |
| w/o SRS | 0.430 | 0.451 | 0.273 | 0.336 | 0.219 | 0.277 | 0.413 | 0.284 |
| w/o Selective Patching | 0.423 | 0.441 | 0.267 | 0.331 | 0.215 | 0.271 | 0.408 | 0.277 |
| w/o Dynamic Reassembly | 0.413 | 0.428 | 0.261 | 0.324 | 0.193 | 0.246 | 0.407 | 0.278 |
| w/o Adaptive Fusion | 0.412 | 0.429 | 0.263 | 0.326 | 0.196 | 0.251 | 0.402 | 0.277 |
| SRSNet | **0.404** | **0.424** | **0.252** | **0.314** | **0.183** | **0.239** | **0.392** | **0.270** |

more potentially useful candidates of representation space. The Adaptive Fusion makes a convex combination of the embeddings from both the conventional patching and SRS, providing complementary enhancement. Additionally, all these components cooperate with each other to construct the SRS Module, producing positive impact on forecasting performance.

## 5.4 Efficiency Analysis

Our proposed SRS, as a modular plugin, mainly consists of two MLPs: $Scorer^s$ and $Scorer^r$. The SRSNet consists of SRS and an MLP layer, is also significantly lightweight compared to other multi-layer stacked neural networks. Table 5–6 report the efficiency of SRSNet the SRS module.

Table 5: Efficiency comparison between SRSNet and other baselines on ETTh1 and Solar datasets with look-back length equals 512, forecasting horizon equals 720, and batch size equals 32. The Max GPU Memory (MB) and Training Time (s) per batch are reported as the main metrics.

| Datasets | | ETTh1 | | Solar | |
|---|---|---|---|---|---|
| Metrics | | Memory (MB) | Training Time (s) | Memory (MB) | Training Time (s) |
| Linear | DLinear | 828 | 1.28 | 815 | 15.66 |
| | Amplifer | 596 | 1.78 | 715 | 14.38 |
| CNN | TimesNet | 2,846 | 14.95 | 13,141 | 1812.46 |
| | FEDformer | 8,190 | 64.83 | 3,751 | 227.45 |
| | Stationary | 18,386 | 40.22 | 18,529 | 156.27 |
| Transformer | Crossformer | 3,976 | 17.13 | 16,375 | 205.60 |
| | PatchTST | 1,404 | 2.49 | 26,777 | 137.60 |
| | iTransformer | 722 | 4.14 | 1,015 | 20.66 |
| | TimeMixer | 1,394 | 7.49 | 20,602 | 107.15 |
| MLP | TimeKAN | 1,456 | 5.50 | 13,109 | 326.38 |
| | SRSNet | 1,012 | 2.27 | 6,301 | 56.149 |

Table 6: Efficiency analysis of the SRS module (the same settings as in Table 5). The Max GPU Memory (MB), Inference Time (s) per batch, Training Time (s) per batch, and Multiply-Accumulate Operations (MACs) are reported as the main metrics.

| Datasets | Variants | Memory (MB) | Inference (s) | Training (s) | MACs (G) |
|---|---|---|---|---|---|
| ETTh1 | PatchTST | 2,837 | 5.076 | 5.131 | 16.214 |
| | +SRS | 2,907 | 5.722 | 5.763 | 16.905 |
| | Overhead | +2.47% | +12.73% | +12.31% | +4.26% |
| | Crossformer | 4,011 | 27.503 | 32.613 | 56.280 |
| | +SRS | 4,159 | 30.311 | 35.276 | 56.625 |
| | Overhead | +3.69% | +10.21% | +8.17% | +0.61% |
| Solar | PatchTST | 27,822.08 | 84.231 | 88.714 | 600.261 |
| | +SRS | 29,767.68 | 95.200 | 101.981 | 613.790 |
| | Overhead | +6.99% | +13.02% | +14.95% | +2.25% |
| | Crossformer | 17,355 | 79.031 | 82.472 | 61.822 |
| | +SRS | 18,978 | 86.674 | 90.268 | 62.174 |
| | Overhead | +9.35% | +9.67% | +9.45% | +0.57% |

The results in Table 5 show that SRSNet has significant advantages when compared with Transformer-based models such as Crossformer, FEDformer, and PatchTST. For iTransformer, though it performs more efficiently due to its embedding of the whole series, it consistently performs worse that SRSNet under all the scenarios. When compared with recent state-of-the-art baselines TimeMixer and TimeKAN, SRSNet also shows better efficiency. Though DLinear and Amplifer are the most efficienct model due to their non-patch design and simple structures, they perform poorly on large datasets such as Solar and Traffic, reflecting an insufficient capability in modeling long-term dependencies and nonlinear learning. In summary, SRSNet achives a good balance between efficiency and performance, demonstrating the SRS module organizes effective representation spaces for pattern learning.

In Table 6, the results further reveal the overhead introduced by the SRS module. Integrated like a plugin into classical patch-based models PatchTST and Crossformer, the SRS module only introduces about 10% increasement of Max GPU Memory, Inference Time, and Training Time, and under 5% increasement of MACs, demonstarting the strong practicality in real-world applications.

## 5.5 Parameter Sensitivity

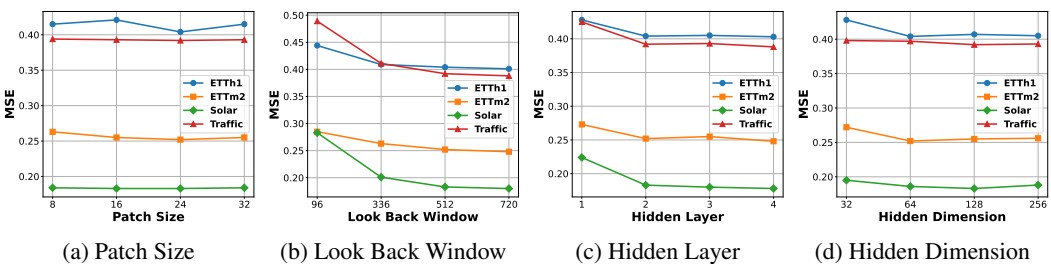

| (a) Patch Size | (b) Look Back Window | (c) Hidden Layer | (d) Hidden Dimension |

Figure 4: Parameter sensitivity studies of main hyper-parameters in SRSNet.

We study the parameter sensitivity of SRSNet. Figure 4a shows the performance of SRSNet under different patch sizes. Since the SRS module adaptively selects useful patches to contruct the representation space, it shows robust performance and we often choose 16 or 24 as common configurations.

Figure 4b demonstrates the capability of SRSNet to fully utilize the historical information. As the length of Look Back Window extends, the performance becomes better. Figure 4c and Figure 4c show the influences of the hidden layer and hidden dimension in the $\text{Scorer}^s$ and $\text{Scorer}^r$ of SRS, which determine the capacity of constructing the Selective Representation Space. The common configurations of $\text{Scorer}^s$ and $\text{Scorer}^r$ are 2 hidden layers with hidden dimension equals 128.

## 6  Discussion

**Potential limitations** The SRS module demonstrates its efficacy on SRSNet and other patch-based models in LTSF scenarios. However, there are several potential limitations of SRS that warrant discussion here:

- **Impractical to non-patch models**: The SRS module may not be suitable for those state-of-the-art models which dose not adopt patching techniques. If it is used compulsorily, it can be regarded as the case where the patch size equals 1. However, this will lose the semantics of the patch itself and cause additional computational complexity overhead, which is not practical. Luckily, current works about expert models and foundation models mainly focus on patch-based modeling, thus providing more application prospects for the SRS module.

- **Scaling law:** We observe that the SRS module works stably with fixed hyperparameters such as the hidden size and layer number on end-to-end scenarios. In the pretrain scenarios, which adopt foundation models with billion-scale time series corpus, the effectiveness and generalization of the SRS module need to be furtherly verified. Whether the SRS module obeys the scaling law also needs more practical evidence.

- **Interpretability:** Since the SRS module is also optimized through gradient descent and the gradient flow is coupled with the subsequent patch-based models, we can only ensure the selected patches are useful for forecasting, but not all of them are interpretable. Therefore, we choose to allow maximum freedom of the SRS module to adaptively dertermine which patches to choose and the order of them.

- **Initialization:** The initialization of the weights $\alpha$ seems to be important. Although a random initialization of $\alpha$ can also lead to best forecasting performance under a few epochs, we recommend to manually initialize it when the prior knowledge of datasets is acquired, such as increasing it when the datasets are periodic and stationary, and decreasing it when the datasets are non-stationary and shifting. This can help enchance the convergence and stability of the SRS module.

**Future work** The current SRS module is purely data-driven, which preserves lightweight but lacks interpretability. In the future, we hope to devise an environment-aware mechanism to perceive the patch-wise data distributions and patterns more explictly, which provides a potential solution for designing the representation spaces adaptively and explainably. To further enhance the practical value of SRS in Time Series Foundation Models, we think its capacity needs to be expanded to memorize and distinguish heterogeneous data patterns, where the Mixture-of-Experts (MoE) with a strong enough router can be a good solution. To solve the initialization problem in the SRS modlue, a more efficient update mechanism for $\alpha$ deserves exploration. A potential solution is to design a module to supervise the sample-wise data patterns, constructing a more explict optimization objective between data patterns and $\alpha$, which can also enhance the interpretability of the SRS module.

## 7  Conclusion

In this paper, we propose a modular SRS mainly composed of the Selective Patching and Dynamic Reassembly, which adaptively select the patches and then reassemble them to fully exploit the information in the contextual time series, thus mitigating the adverse effects from special phenomenons like changable periods, anomalies, and shifting. The SRS module is gradient-based and works in a plug-and-play paradigm to enhance the patch-based models in forecasting tasks. We also proposes a simple yet effective SRSNet composed of the SRS module and an MLP head. Comprehensive experiments on real-world datasets demonstrate that SRSNet achieves state-of-the-art performance, and the SRS module can effectively improve the performance of patch-based models.

## Acknowledgments and Disclosure of Funding

This work was partially supported by the National Natural Science Foundation of China (62472174, 62372179), and ECNU Multifunctional Platform for Innovation (001). Bin Yang is the corresponding author of the work.

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

# A Experimental Details

Table 7: Statistics of datasets.

| Dataset | Domain | Frequency | Lengths | Dim | Split | Description |
|---|---|---|---|---|---|---|
| ETTh1 | Electricity | 1 hour | 14,400 | 7 | 6:2:2 | Power transformer 1, comprising seven indicators such as oil temperature and useful load |
| ETTh2 | Electricity | 1 hour | 14,400 | 7 | 6:2:2 | Power transformer 2, comprising seven indicators such as oil temperature and useful load |
| ETTm1 | Electricity | 15 mins | 57,600 | 7 | 6:2:2 | Power transformer 1, comprising seven indicators such as oil temperature and useful load |
| ETTm2 | Electricity | 15 mins | 57,600 | 7 | 6:2:2 | Power transformer 2, comprising seven indicators such as oil temperature and useful load |
| Weather | Environment | 10 mins | 52,696 | 21 | 7:1:2 | Recorded every for the whole year 2020, which contains 21 meteorological indicators |
| Electricity | Electricity | 1 hour | 26,304 | 321 | 7:1:2 | Electricity records the electricity consumption in kWh every 1 hour from 2012 to 2014 |
| Solar | Energy | 10 mins | 52,560 | 137 | 6:2:2 | Solar production records collected from 137 PV plants in Alabama |
| Traffic | Traffic | 1 hour | 17,544 | 862 | 7:1:2 | Road occupancy rates measured by 862 sensors on San Francisco Bay area freeways |

We utilize the TFB code repository for unified evaluation, with all baseline results also derived from TFB. Following the settings in TFB Qiu et al. [2024], we do not apply the "Drop Last" trick to ensure a fair comparison. All experiments of SRSNet are conducted using PyTorch Paszke et al. [2019] in Python 3.8 and executed on an NVIDIA Tesla-A800 GPU. The training process is guided by the MSE loss function and employs the ADAM optimizer. The initial batch size is set to 64, with the flexibility to halve it (down to a minimum of 8) in case of an Out-Of-Memory (OOM) issue.

## A.1 Full results

Table 8: Multivariate forecasting results with forecasting horizons $F \in \{96, 192, 336, 720\}$ for the datasets. **Red**: the best, **Blue**: the 2nd best. The standard deviation of SRSNet calculated through 5 random seeds are also reported.

| Models | | SRSNet (ours) | | TimeKAN (2025) | | Amplifier (2025) | | iTransformer (2024) | | TimeMixer (2024) | | PatchTST (2023) | | Crossformer (2023) | | TimesNet (2023) | | DLinear (2023) | | Stationary (2022) | | FEDformer (2022) | |
|---|---|---|---|---|---|---|---|---|---|---|---|---|---|---|---|---|---|---|---|---|---|---|---|
| Metrics | | mse | mae | mse | mae | mse | mae | mse | mae | mse | mae | mse | mae | mse | mae | mse | mae | mse | mae | mse | mae | mse | mae |
| ETTh1 | 96 | 0.366±0.001 | 0.394±0.001 | 0.370 | 0.396 | 0.373 | 0.399 | 0.386 | 0.405 | 0.372 | 0.401 | 0.377 | 0.397 | 0.411 | 0.435 | 0.389 | 0.412 | 0.379 | 0.403 | 0.591 | 0.524 | 0.379 | 0.419 |
| | 192 | 0.400±0.001 | 0.415±0.001 | 0.403 | 0.417 | 0.414 | 0.420 | 0.430 | 0.435 | 0.413 | 0.430 | 0.409 | 0.425 | 0.409 | 0.438 | 0.440 | 0.443 | 0.427 | 0.435 | 0.615 | 0.540 | 0.420 | 0.444 |
| | 336 | 0.424±0.002 | 0.430±0.001 | 0.420 | 0.432 | 0.442 | 0.446 | 0.450 | 0.452 | 0.438 | 0.450 | 0.431 | 0.444 | 0.433 | 0.457 | 0.523 | 0.487 | 0.440 | 0.440 | 0.632 | 0.551 | 0.458 | 0.466 |
| | 720 | 0.426±0.001 | 0.455±0.001 | 0.442 | 0.463 | 0.455 | 0.467 | 0.495 | 0.487 | 0.483 | 0.483 | 0.457 | 0.477 | 0.501 | 0.514 | 0.521 | 0.495 | 0.473 | 0.494 | 0.828 | 0.658 | 0.474 | 0.488 |
| ETTh2 | 96 | 0.271±0.002 | 0.338±0.001 | 0.287 | 0.343 | 0.292 | 0.347 | 0.292 | 0.347 | 0.270 | 0.342 | 0.274 | 0.337 | 0.728 | 0.603 | 0.334 | 0.370 | 0.300 | 0.364 | 0.347 | 0.387 | 0.337 | 0.380 |
| | 192 | 0.324±0.001 | 0.372±0.001 | 0.329 | 0.382 | 0.348 | 0.393 | 0.348 | 0.384 | 0.349 | 0.387 | 0.348 | 0.384 | 0.723 | 0.607 | 0.404 | 0.413 | 0.387 | 0.423 | 0.379 | 0.418 | 0.415 | 0.428 |
| | 336 | 0.349±0.001 | 0.394±0.003 | 0.370 | 0.412 | 0.383 | 0.423 | 0.372 | 0.407 | 0.367 | 0.410 | 0.377 | 0.416 | 0.740 | 0.628 | 0.389 | 0.435 | 0.490 | 0.487 | 0.358 | 0.413 | 0.389 | 0.457 |
| | 720 | 0.391±0.002 | 0.434±0.002 | 0.420 | 0.450 | 0.407 | 0.444 | 0.424 | 0.444 | 0.401 | 0.436 | 0.406 | 0.441 | 1.386 | 0.882 | 0.434 | 0.448 | 0.704 | 0.597 | 0.422 | 0.457 | 0.483 | 0.488 |
| ETTm1 | 96 | 0.288±0.002 | 0.341±0.002 | 0.290 | 0.348 | 0.292 | 0.346 | 0.287 | 0.342 | 0.293 | 0.345 | 0.289 | 0.343 | 0.314 | 0.367 | 0.340 | 0.378 | 0.300 | 0.345 | 0.415 | 0.410 | 0.463 | 0.463 |
| | 192 | 0.326±0.001 | 0.364±0.002 | 0.332 | 0.368 | 0.327 | 0.365 | 0.331 | 0.371 | 0.335 | 0.372 | 0.329 | 0.368 | 0.374 | 0.410 | 0.392 | 0.404 | 0.336 | 0.366 | 0.494 | 0.451 | 0.575 | 0.516 |
| | 336 | 0.365±0.003 | 0.386±0.001 | 0.354 | 0.386 | 0.365 | 0.386 | 0.358 | 0.384 | 0.368 | 0.386 | 0.362 | 0.390 | 0.413 | 0.432 | 0.423 | 0.426 | 0.367 | 0.386 | 0.577 | 0.490 | 0.618 | 0.544 |
| | 720 | 0.426±0.001 | 0.420±0.002 | 0.401 | 0.417 | 0.427 | 0.419 | 0.412 | 0.416 | 0.426 | 0.417 | 0.416 | 0.423 | 0.753 | 0.613 | 0.475 | 0.453 | 0.419 | 0.416 | 0.636 | 0.535 | 0.612 | 0.551 |
| ETTm2 | 96 | 0.164±0.001 | 0.254±0.001 | 0.164 | 0.254 | 0.164 | 0.254 | 0.168 | 0.262 | 0.165 | 0.256 | 0.165 | 0.255 | 0.296 | 0.391 | 0.189 | 0.265 | 0.164 | 0.255 | 0.210 | 0.294 | 0.216 | 0.309 |
| | 192 | 0.220±0.001 | 0.296±0.001 | 0.238 | 0.300 | 0.226 | 0.300 | 0.224 | 0.295 | 0.225 | 0.298 | 0.221 | 0.293 | 0.369 | 0.416 | 0.254 | 0.310 | 0.224 | 0.304 | 0.338 | 0.373 | 0.297 | 0.360 |
| | 336 | 0.271±0.001 | 0.327±0.001 | 0.278 | 0.331 | 0.276 | 0.331 | 0.274 | 0.330 | 0.277 | 0.332 | 0.276 | 0.327 | 0.588 | 0.600 | 0.313 | 0.345 | 0.277 | 0.337 | 0.432 | 0.416 | 0.366 | 0.400 |
| | 720 | 0.353±0.001 | 0.380±0.002 | 0.359 | 0.387 | 0.358 | 0.388 | 0.367 | 0.385 | 0.360 | 0.387 | 0.362 | 0.381 | 0.750 | 0.612 | 0.413 | 0.402 | 0.371 | 0.401 | 0.554 | 0.476 | 0.459 | 0.450 |
| Weather | 96 | 0.147±0.001 | 0.198±0.001 | 0.151 | 0.202 | 0.147 | 0.199 | 0.157 | 0.207 | 0.147 | 0.198 | 0.150 | 0.200 | 0.143 | 0.210 | 0.168 | 0.214 | 0.170 | 0.230 | 0.188 | 0.242 | 0.229 | 0.298 |
| | 192 | 0.190±0.001 | 0.242±0.001 | 0.195 | 0.244 | 0.194 | 0.245 | 0.200 | 0.248 | 0.191 | 0.242 | 0.191 | 0.239 | 0.195 | 0.261 | 0.219 | 0.262 | 0.216 | 0.275 | 0.240 | 0.290 | 0.265 | 0.334 |
| | 336 | 0.241±0.001 | 0.282±0.001 | 0.242 | 0.287 | 0.243 | 0.282 | 0.252 | 0.287 | 0.244 | 0.280 | 0.242 | 0.279 | 0.254 | 0.319 | 0.278 | 0.302 | 0.258 | 0.307 | 0.322 | 0.328 | 0.330 | 0.372 |
| | 720 | 0.325±0.001 | 0.340±0.001 | 0.317 | 0.340 | 0.310 | 0.329 | 0.320 | 0.336 | 0.316 | 0.331 | 0.312 | 0.330 | 0.335 | 0.385 | 0.353 | 0.351 | 0.324 | 0.367 | 0.396 | 0.378 | 0.423 | 0.418 |
| Electricity | 96 | 0.131±0.001 | 0.226±0.001 | 0.135 | 0.231 | 0.132 | 0.227 | 0.134 | 0.230 | 0.153 | 0.256 | 0.143 | 0.247 | 0.134 | 0.231 | 0.169 | 0.271 | 0.140 | 0.237 | 0.171 | 0.274 | 0.191 | 0.305 |
| | 192 | 0.147±0.002 | 0.241±0.001 | 0.149 | 0.243 | 0.149 | 0.243 | 0.154 | 0.250 | 0.168 | 0.269 | 0.158 | 0.260 | 0.146 | 0.243 | 0.180 | 0.280 | 0.154 | 0.250 | 0.180 | 0.283 | 0.203 | 0.316 |
| | 336 | 0.163±0.001 | 0.258±0.001 | 0.165 | 0.260 | 0.167 | 0.261 | 0.169 | 0.265 | 0.189 | 0.291 | 0.168 | 0.267 | 0.165 | 0.264 | 0.204 | 0.293 | 0.169 | 0.268 | 0.204 | 0.305 | 0.221 | 0.333 |
| | 720 | 0.201±0.001 | 0.291±0.001 | 0.206 | 0.297 | 0.203 | 0.292 | 0.194 | 0.288 | 0.228 | 0.320 | 0.214 | 0.307 | 0.237 | 0.314 | 0.206 | 0.293 | 0.203 | 0.300 | 0.221 | 0.319 | 0.259 | 0.364 |
| Solar | 96 | 0.167±0.002 | 0.222±0.001 | 0.187 | 0.255 | 0.175 | 0.237 | 0.174 | 0.229 | 0.180 | 0.233 | 0.170 | 0.234 | 0.183 | 0.208 | 0.198 | 0.270 | 0.199 | 0.265 | 0.365 | 0.390 | 0.485 | 0.570 |
| | 192 | 0.182±0.003 | 0.237±0.001 | 0.194 | 0.265 | 0.198 | 0.259 | 0.205 | 0.270 | 0.201 | 0.259 | 0.204 | 0.302 | 0.208 | 0.226 | 0.206 | 0.276 | 0.220 | 0.282 | 0.400 | 0.386 | 0.415 | 0.477 |
| | 336 | 0.188±0.002 | 0.245±0.003 | 0.203 | 0.264 | 0.213 | 0.259 | 0.216 | 0.282 | 0.214 | 0.272 | 0.212 | 0.293 | 0.212 | 0.239 | 0.208 | 0.284 | 0.234 | 0.295 | 0.414 | 0.394 | 1.008 | 0.839 |
| | 720 | 0.195±0.002 | 0.251±0.002 | 0.209 | 0.269 | 0.222 | 0.269 | 0.211 | 0.260 | 0.218 | 0.278 | 0.215 | 0.307 | 0.215 | 0.256 | 0.232 | 0.294 | 0.243 | 0.301 | 0.379 | 0.377 | 0.655 | 0.627 |
| Traffic | 96 | 0.361±0.001 | 0.254±0.001 | 0.388 | 0.269 | 0.391 | 0.277 | 0.363 | 0.265 | 0.369 | 0.256 | 0.370 | 0.262 | 0.526 | 0.288 | 0.595 | 0.312 | 0.395 | 0.275 | 0.603 | 0.330 | 0.593 | 0.365 |
| | 192 | 0.380±0.001 | 0.263±0.001 | 0.411 | 0.286 | 0.405 | 0.283 | 0.384 | 0.273 | 0.400 | 0.271 | 0.386 | 0.269 | 0.503 | 0.263 | 0.613 | 0.322 | 0.407 | 0.280 | 0.611 | 0.338 | 0.614 | 0.381 |
| | 336 | 0.392±0.001 | 0.270±0.001 | 0.425 | 0.284 | 0.416 | 0.290 | 0.396 | 0.277 | 0.412 | 0.272 | 0.396 | 0.275 | 0.505 | 0.276 | 0.626 | 0.332 | 0.417 | 0.286 | 0.628 | 0.342 | 0.627 | 0.389 |
| | 720 | 0.434±0.001 | 0.293±0.001 | 0.455 | 0.302 | 0.454 | 0.312 | 0.445 | 0.308 | 0.462 | 0.316 | 0.435 | 0.295 | 0.552 | 0.301 | 0.635 | 0.340 | 0.454 | 0.308 | 0.646 | 0.350 | 0.646 | 0.394 |
| 1st Count | | 23 | 20 | 3 | 0 | 1 | 2 | 2 | 3 | 1 | 0 | 0 | 4 | 2 | 3 | 0 | 0 | 0 | 0 | 0 | 0 | 0 | 0 |

Table 9: Ablation Studies with forecasting horizons $F \in \{96, 192, 336, 720\}$ for the datasets. **Red**: the best.

| Datasets | ETTh1 | | | | | | | | ETTm2 | | | | | | | |
|---|---|---|---|---|---|---|---|---|---|---|---|---|---|---|---|---|
| Horizons | 96 | | 192 | | 336 | | 720 | | 96 | | 192 | | 336 | | 720 | |
| Metrics | MSE | MAE | MSE | MAE | MSE | MAE | MSE | MAE | MSE | MAE | MSE | MAE | MSE | MAE | MSE | MAE |
| w/o SRS | 0.385 | 0.412 | 0.427 | 0.448 | 0.447 | 0.461 | 0.462 | 0.483 | 0.178 | 0.272 | 0.243 | 0.316 | 0.291 | 0.352 | 0.378 | 0.402 |
| w/o Selective Patching | 0.381 | 0.407 | 0.418 | 0.428 | 0.441 | 0.457 | 0.452 | 0.471 | 0.177 | 0.271 | 0.229 | 0.304 | 0.290 | 0.353 | 0.372 | 0.395 |
| w/o Dynamic Reassembly | 0.373 | 0.398 | 0.406 | 0.417 | 0.429 | 0.436 | 0.443 | 0.462 | 0.169 | 0.263 | 0.231 | 0.306 | 0.275 | 0.335 | 0.368 | 0.391 |
| w/o Adaptive Fusion | 0.371 | 0.397 | 0.411 | 0.423 | 0.428 | 0.434 | 0.437 | 0.461 | 0.172 | 0.266 | 0.237 | 0.311 | 0.278 | 0.339 | 0.365 | 0.388 |
| SRSNet | 0.366 | 0.394 | 0.400 | 0.415 | 0.424 | 0.430 | 0.426 | 0.455 | 0.164 | 0.254 | 0.220 | 0.296 | 0.271 | 0.328 | 0.353 | 0.380 |

Table 10: Ablation Studies with forecasting horizons $F \in \{96, 192, 336, 720\}$ for the datasets. **Red**: the best.

| Datasets | Solar | | | | | | | | Traffic | | | | | | | |
|---|---|---|---|---|---|---|---|---|---|---|---|---|---|---|---|---|
| Horizons | 96 | | 192 | | 336 | | 720 | | 96 | | 192 | | 336 | | 720 | |
| Metrics | MSE | MAE | MSE | MAE | MSE | MAE | MSE | MAE | MSE | MAE | MSE | MAE | MSE | MAE | MSE | MAE |
| w/o SRS | 0.192 | 0.243 | 0.215 | 0.271 | 0.228 | 0.292 | 0.239 | 0.301 | 0.387 | 0.272 | 0.402 | 0.275 | 0.412 | 0.284 | 0.452 | 0.304 |
| w/o Selective Patching | 0.189 | 0.241 | 0.209 | 0.254 | 0.223 | 0.286 | 0.237 | 0.304 | 0.385 | 0.268 | 0.399 | 0.272 | 0.410 | 0.273 | 0.437 | 0.296 |
| w/o Dynamic Reassembly | 0.186 | 0.227 | 0.196 | 0.241 | 0.192 | 0.253 | 0.199 | 0.262 | 0.381 | 0.264 | 0.403 | 0.277 | 0.398 | 0.274 | 0.446 | 0.298 |
| w/o Adaptive Fusion | 0.182 | 0.226 | 0.195 | 0.239 | 0.198 | 0.262 | 0.207 | 0.278 | 0.378 | 0.262 | 0.389 | 0.269 | 0.399 | 0.276 | 0.443 | 0.301 |
| SRSNet | **0.167** | **0.222** | **0.182** | **0.237** | **0.188** | **0.245** | **0.195** | **0.251** | **0.361** | **0.254** | **0.380** | **0.263** | **0.392** | **0.270** | **0.434** | **0.293** |

Table 11: Plugin Studies with forecasting horizons $F \in \{96, 192, 336, 720\}$ for the datasets. **Black**: the improvement.

| Datasets | ETTh1 | | | | | | | | ETTm2 | | | | | | | |
|---|---|---|---|---|---|---|---|---|---|---|---|---|---|---|---|---|
| Horizons | 96 | | 192 | | 336 | | 720 | | 96 | | 192 | | 336 | | 720 | |
| Metrics | MSE | MAE | MSE | MAE | MSE | MAE | MSE | MAE | MSE | MAE | MSE | MAE | MSE | MAE | MSE | MAE |
| MLP | 0.385 | 0.412 | 0.427 | 0.448 | 0.447 | 0.461 | 0.462 | 0.483 | 0.178 | 0.272 | 0.243 | 0.316 | 0.291 | 0.352 | 0.378 | 0.402 |
| + SRS | 0.366 | 0.394 | 0.400 | 0.415 | 0.424 | 0.430 | 0.426 | 0.455 | 0.164 | 0.254 | 0.220 | 0.296 | 0.271 | 0.328 | 0.353 | 0.380 |
| Improve | 4.94% | 4.37% | 6.32% | 7.37% | 5.15% | 6.72% | 7.79% | 5.80% | 7.87% | 6.62% | 9.47% | 6.33% | 6.87% | 6.82% | 6.61% | 5.47% |
| PatchTST | 0.377 | 0.397 | 0.409 | 0.425 | 0.431 | 0.444 | 0.457 | 0.477 | 0.165 | 0.255 | 0.221 | 0.293 | 0.276 | 0.327 | 0.362 | 0.381 |
| + SRS | 0.364 | 0.393 | 0.398 | 0.419 | 0.422 | 0.438 | 0.431 | 0.453 | 0.159 | 0.248 | 0.213 | 0.284 | 0.266 | 0.319 | 0.358 | 0.377 |
| Improve | 3.45% | 1.01% | 2.69% | 1.41% | 2.09% | 1.35% | 5.69% | 5.03% | 3.64% | 2.75% | 3.62% | 3.07% | 3.62% | 2.45% | 1.10% | 1.05% |
| Crossformer | 0.411 | 0.435 | 0.409 | 0.438 | 0.433 | 0.457 | 0.501 | 0.514 | 0.296 | 0.391 | 0.369 | 0.416 | 0.588 | 0.600 | 0.750 | 0.612 |
| + SRS | 0.401 | 0.423 | 0.404 | 0.432 | 0.428 | 0.452 | 0.494 | 0.511 | 0.276 | 0.356 | 0.324 | 0.396 | 0.484 | 0.496 | 0.730 | 0.601 |
| Improve | 2.43% | 2.76% | 1.22% | 1.37% | 1.15% | 1.09% | 1.40% | 0.58% | 6.76% | 8.95% | 12.20% | 4.81% | 17.69% | 17.33% | 2.67% | 1.80% |
| PatchMLP | 0.380 | 0.395 | 0.430 | 0.441 | 0.451 | 0.453 | 0.479 | 0.484 | 0.168 | 0.259 | 0.228 | 0.300 | 0.275 | 0.330 | 0.371 | 0.398 |
| + SRS | 0.378 | 0.393 | 0.419 | 0.430 | 0.428 | 0.447 | 0.461 | 0.475 | 0.161 | 0.251 | 0.222 | 0.294 | 0.268 | 0.323 | 0.362 | 0.390 |
| Improve | 0.53% | 0.51% | 2.56% | 2.49% | 5.10% | 1.32% | 3.76% | 1.86% | 4.17% | 3.09% | 2.63% | 2.00% | 2.55% | 2.12% | 2.43% | 2.01% |
| xPatch | 0.368 | 0.396 | 0.408 | 0.421 | 0.436 | 0.435 | 0.453 | 0.465 | 0.160 | 0.245 | 0.219 | 0.287 | 0.272 | 0.323 | 0.361 | 0.378 |
| + SRS | 0.359 | 0.389 | 0.402 | 0.417 | 0.428 | 0.431 | 0.436 | 0.451 | 0.157 | 0.241 | 0.209 | 0.281 | 0.261 | 0.317 | 0.347 | 0.371 |
| Improve | 2.45% | 1.77% | 1.47% | 0.95% | 1.83% | 0.92% | 3.75% | 3.01% | 1.88% | 1.63% | 4.57% | 2.09% | 4.04% | 1.86% | 3.88% | 1.85% |

Table 12: Plugin Studies with forecasting horizons $F \in \{96, 192, 336, 720\}$ for the datasets. **Black**: the Improvement.

| Datasets | Solar | | | | | | | | Traffic | | | | | | | |
|---|---|---|---|---|---|---|---|---|---|---|---|---|---|---|---|---|
| Horizons | 96 | | 192 | | 336 | | 720 | | 96 | | 192 | | 336 | | 720 | |
| Metrics | MSE | MAE | MSE | MAE | MSE | MAE | MSE | MAE | MSE | MAE | MSE | MAE | MSE | MAE | MSE | MAE |
| MLP | 0.192 | 0.243 | 0.215 | 0.271 | 0.228 | 0.292 | 0.239 | 0.301 | 0.387 | 0.272 | 0.402 | 0.275 | 0.412 | 0.284 | 0.452 | 0.304 |
| + SRS | 0.167 | 0.222 | 0.182 | 0.237 | 0.188 | 0.245 | 0.195 | 0.251 | 0.361 | 0.254 | 0.380 | 0.263 | 0.392 | 0.270 | 0.434 | 0.293 |
| Improve | 13.02% | 8.64% | 15.35% | 12.55% | 17.54% | 16.10% | 18.41% | 16.61% | 6.72% | 6.62% | 5.47% | 4.36% | 4.85% | 4.93% | 3.98% | 3.62% |
| PatchTST | 0.170 | 0.234 | 0.204 | 0.302 | 0.212 | 0.293 | 0.215 | 0.307 | 0.370 | 0.262 | 0.386 | 0.269 | 0.396 | 0.275 | 0.435 | 0.295 |
| + SRS | 0.164 | 0.226 | 0.180 | 0.249 | 0.181 | 0.246 | 0.203 | 0.284 | 0.352 | 0.247 | 0.373 | 0.258 | 0.387 | 0.268 | 0.433 | 0.292 |
| Improve | 3.53% | 3.42% | 11.76% | 17.55% | 14.62% | 16.04% | 5.58% | 7.49% | 4.86% | 5.73% | 3.37% | 4.09% | 2.27% | 2.55% | 0.46% | 1.02% |
| Crossformer | 0.183 | 0.208 | 0.208 | 0.226 | 0.212 | 0.239 | 0.215 | 0.256 | 0.526 | 0.288 | 0.503 | 0.263 | 0.505 | 0.276 | 0.552 | 0.301 |
| + SRS | 0.176 | 0.203 | 0.196 | 0.222 | 0.206 | 0.236 | 0.193 | 0.240 | 0.518 | 0.279 | 0.493 | 0.255 | 0.499 | 0.269 | 0.537 | 0.293 |
| Improve | 3.83% | 2.40% | 5.77% | 1.77% | 2.83% | 1.26% | 10.23% | 6.25% | 1.52% | 3.12% | 1.99% | 3.04% | 1.19% | 2.54% | 2.72% | 2.66% |
| PatchMLP | 0.173 | 0.234 | 0.192 | 0.247 | 0.199 | 0.255 | 0.208 | 0.264 | 0.388 | 0.273 | 0.399 | 0.277 | 0.412 | 0.286 | 0.452 | 0.311 |
| + SRS | 0.163 | 0.222 | 0.178 | 0.244 | 0.183 | 0.247 | 0.193 | 0.256 | 0.381 | 0.264 | 0.389 | 0.268 | 0.400 | 0.278 | 0.439 | 0.298 |
| Improve | 5.64% | 5.05% | 7.19% | 1.12% | 7.92% | 3.14% | 7.26% | 3.14% | 1.86% | 3.36% | 2.51% | 3.08% | 2.98% | 2.93% | 2.92% | 4.25% |
| xPatch | 0.169 | 0.194 | 0.184 | 0.208 | 0.191 | 0.213 | 0.201 | 0.223 | 0.368 | 0.236 | 0.387 | 0.244 | 0.395 | 0.247 | 0.441 | 0.266 |
| + SRS | 0.162 | 0.191 | 0.179 | 0.203 | 0.182 | 0.203 | 0.192 | 0.219 | 0.360 | 0.233 | 0.381 | 0.238 | 0.384 | 0.232 | 0.431 | 0.256 |
| Improve | 4.14% | 1.55% | 2.72% | 2.40% | 4.71% | 4.69% | 4.48% | 1.79% | 2.11% | 1.42% | 1.53% | 2.28% | 2.78% | 6.07% | 2.27% | 3.76% |

## A.2 Showcases

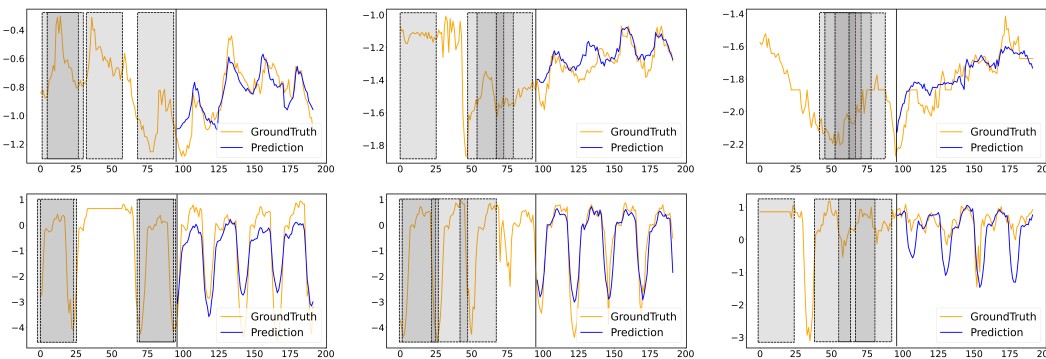

Figure 5: Visualization of input-96-predict-96 results on the ETTh1 dataset. SRSNet effectively processes the special cases with the help of SRS module. The grey rectangles are the selected patches with the size of 24.

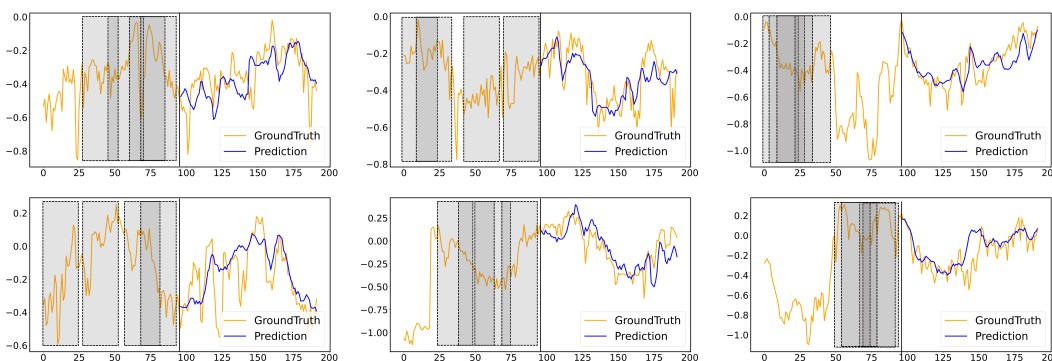

Figure 6: Visualization of input-96-predict-96 results on the ETTm2 dataset. SRSNet effectively processes the special cases with the help of SRS module. The grey rectangles are the selected patches with the size of 24.

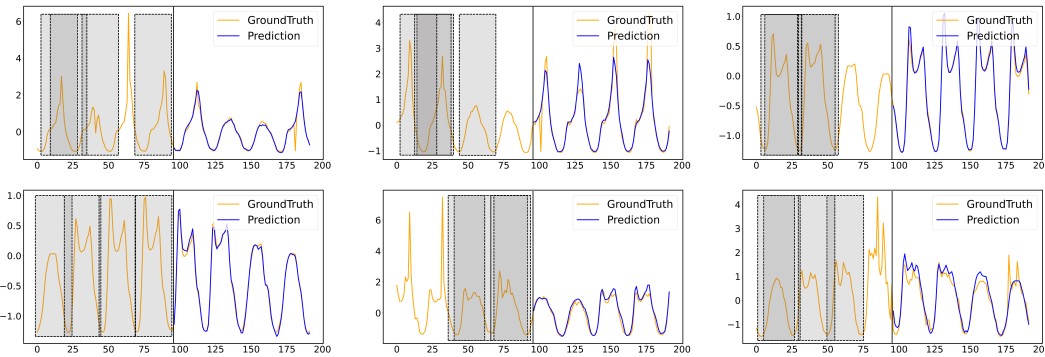

Figure 7: Visualization of input-96-predict-96 results on the Traffic dataset. SRSNet effectively processes the special cases with the help of SRS module. The grey rectangles are the selected patches with the size of 24.

## B  Related Works

Time Series Analysis is very important in many fields like economy [Qiu et al., 2025b,c, Liu et al., 2025a, Wu et al., 2023b], transportation [Wu et al., 2024a,b, Liu et al., 2025b, Lu et al., 2011, Pan et al., 2023], health [Lu et al., 2023, 2024a,b], weather [Li et al., 2025a, Yang et al., 2024, Zhou et al.,

2025, Cheng et al., 2023], energy [Li et al., 2025b, Wang et al., 2025a, Wu et al., 2025, Guo et al., 2014], including multiple key tasks like forecasting [Qiu et al., 2025d, Dai et al., 2024a, Hu et al., 2025a, Dai et al., 2024b, Liu et al., 2025c], anomaly detection [Shentu et al., 2025, Li et al., 2025c, Qiu et al., 2025e, Wu et al., 2024c], classification [Wu et al., 2023a, Wang et al., 2024, Liu et al., 2024], imputation [Yao et al., 2024a], and others [Qiu et al., 2025f, Yao et al., 2024b,c]. Among them, Time Series Forecasting is most widely used in real-world applications.

Time series forecasting (TSF) [Sun et al., 2025b, Yao et al., 2023, Liu et al., 2025d] ] involves the prediction of future observations grounded in historical ones. Research findings have indicated that features derived through learning processes may exhibit superior performance compared to human-engineered features [Wang et al., 2025b,c, 2023, Li et al., 2025d, Yue et al., 2025a, 2024, 2025b, Wang et al., 2025d,e, Ma et al., 2025a,b,c,d,e,f, Huang et al., 2025b,c, 2023, 2024]. By capitalizing on the representation learning capabilities of deep neural networks (DNNs), numerous deep-learning approaches have come into existence. Methods such as TimesNet [Wu et al., 2023a] and SegRNN [Lin et al., 2023] model time series as vector sequences, utilizing convolutional neural networks (CNNs) or recurrent neural networks (RNNs) to capture temporal dependencies. Moreover, Transformer architectures, including Informer [Zhou et al., 2021], TimeFilter [Hu et al., 2025b], TimeBridge [Liu et al., 2025e], PDF [Dai et al., 2024a], Triformer [Cirstea et al., 2022a], PatchTST [Nie et al., 2023], ROSE [Wang et al., 2025f], and LightGTS [Wang et al., 2025g], are capable of more accurately capturing the intricate relationships between time points, thereby significantly enhancing forecasting performance. MLP-based methods, including DUET [Qiu et al., 2025a], AMD [Hu et al., 2025a], SparseTSF [Lin et al., 2024a], and CycleNet [Lin et al., 2024b], adopt relatively simpler architectures with fewer parameters yet still attain highly competitive forecasting accuracy.

