# OpenReview forum: "Enhancing Time Series Forecasting through Selective Representation Spaces: A Patch Perspective"
_NeurIPS.cc/2025/Conference — NeurIPS 2025 spotlight_

### Official Review · Reviewer_Gwnu · 2025-06-23

**Clarity:** 3
**Significance:** 3
**Originality:** 3
**Rating:** 5
**Confidence:** 3

**Summary:**

This work explores a Selective Representation Space (SRS) module, using Selective Patching and Dynamic Reassembly to adaptively select and shuffle patches for better context exploitation. The SRSNet, combining SRS and an MLP head, achieves state-of-the-art results on multi-domain datasets. As a plug-and-play module, SRS also boosts existing patch-based models.

**Questions:**

See the Weaknesses

**Ethical Concerns:**

["NO or VERY MINOR ethics concerns only"]

**Final Justification:**

The author's rebuttal, especially the additional experiments, addressed all of my concerns. Taking all the reviewers' comments into account, I believe this work is solid. I support its acceptance.

**Limitations:**

Yes

**Paper Formatting Concerns:**

No problem

**Quality:**

3

**Strengths And Weaknesses:**

Strengths:

1. The SRS module is a plugin which can enhance the time series forecasting, providing general improvement on patch-based models.

2. The simple baseline SRSNet also demonstrates competitive performance against current strong baselines.

Weaknesses:
1. Please elaborate the definition or concept of Selective Representation Space in more detail and clearly. What is the main difference between patches and representations?

2. Though authors have considered the SRS as a simple baseline or as a plugin, the datasets used are not enough. The "ETT long term forecasting benchmark" is often criticized for its flaws such as limited domain coverage and the practice of forecasting at unreasonable horizons (e.g., 720 days into the future for exchange rate or oil temperature of a transformer at a specific hour months into the future). Every new model somehow beats this benchmark; however, there is still barely any absolute progress, only an illusion of it. Please refer to the talk (and paper) from Christoph Bergmeir [1, 2] where he discusses the limitation of this benchmark and current evaluation practices. A very recent position paper [3] also conducted a comprehensive evaluation of models on this benchmark showing that there's no obvious winner. One (not so difficult) way to improve the quality of evaluation is to include results on better benchmarks [4].

[1] https://neurips.cc/virtual/2024/workshop/84712#collapse108471

[2] Hewamalage, Hansika, Klaus Ackermann, and Christoph Bergmeir. "Forecast evaluation for data scientists: common pitfalls and best practices." Data Mining and Knowledge Discovery 37.2 (2023): 788-832.

[3] Brigato, Lorenzo, et al. "Position: There are no Champions in Long-Term Time Series Forecasting." arXiv preprint arXiv:2502.14045 (2025).

[4] Qiu X, Hu J, Zhou L, et al. "TFB: Towards Comprehensive and Fair Benchmarking of Time Series Forecasting Methods." Proceedings of the VLDB Endowment, 2024, 17(9): 2363-2377.

3. Why are the datasets used in Table 4 corresponding to Efficiency Analysis inconsistent with those in Table 2 and Table 3? Can you provide more details regarding efficiency?

4. I'm still confused about the mechanism of gradient propagation. Please explain the details of formulas (3)-(5).

---

> ### Author Rebuttal · Authors · 2025-07-30
>
> **Reply to W1**
>
> Thanks for the question. In time series forecasting, all the patch-based models obey the calculation chain: Patching —> Embedding (preliminary representations) —> Modeling (senior representations) —> Forecasting (Dimensionality Reduction). Specifically, Patching technique does not directly produce representations, but it determines where the representations come from (the candidate patches). Therefore, we think that the Patching technique actually determines the structure of the representational space, thus affecting the representations. In terms of representations, we think the Embeddings are the preliminary representations directly produced from the patches, and the hidden representations across model layers are the senior ones.
>
> **Reply to W2**
>
> Thanks for the suggestion. To provide more empircal evidence, we conduct additional experiments on another 6 datasets from TFB. Their statistical information is as follows:
>
> | datasets | NASDAQ | NYSE | AQShunyi |
> | --- | --- | --- | --- |
> | Shifting Values | 0.932 (Strong) | 0.620 (Modest) |     0.019 (Weak) |
>
> | datasets | PEMS08 | ZafNoo | AQWan |
> | --- | --- | --- | --- |
> | Seasonality Values | 0.850 (Strong) | 0.757 (Modest) | 0.119 (Weak) |
>
> To comprehensively evaluate the capability of SRS, we make plugin experiments with all 5 models in our paper on the above 6 datasets, the mean results of 4 forecasting horizons are reported as follows:
>
> | Datasets | NASDAQ |  | NYSE |  | AQShunyi |  |
> | --- | --- | --- | --- | --- | --- | --- |
> | Metrics | MSE | MAE | MSE | MAE | MSE | MAE |
> | PatchTST | 0.972 | 0.721 | 0.483 | 0.437 | 0.703 | 0.507 |
> | + SRS | 0.757 | 0.569 | 0.418 | 0.379 | 0.641 | 0.461 |
> | Improved | **21.99%** | **21.03%** | **13.50%** | **13.50%** | **8.83%** | **9.14%** |
> | Crossformer | 1.737 | 0.992 | 0.983 | 0.909 | 0.694 | 0.504 |
> | + SRS | 1.256 | 0.709 | 0.854 | 0.785 | 0.635 | 0.455 |
> | Improved | **27.35%** | **28.10%** | **13.48%** | **13.84%** | **8.64%** | **9.71%** |
> | PatchMLP | 1.352 | 0.810 | 0.467 | 0.440 | 0.722 | 0.516 |
> | + SRS | 1.037 | 0.606 | 0.406 | 0.380 | 0.666 | 0.475 |
> | Improved | **23.31%** | **25.30%** | **13.77%** | **13.82%** | **7.78%** | **8.04%** |
> | xPatch | 1.060 | 0.708 | 0.545 | 0.474 | 0.735 | 0.509 |
> | + SRS | 0.850 | 0.563 | 0.469 | 0.408 | 0.678 | 0.464 |
> | Improved | **19.63%** | **20.32%** | **14.56%** | **14.63%** | **7.68%** | **8.75%** |
> | MLP | 1.200 | 0.803 | 0.463 | 0.432 | 0.717 | 0.513 |
> | + SRS | 0.925 | 0.600 | 0.395 | 0.366 | 0.662 | 0.467 |
> | Improved | **23.50%** | **25.50%** | **15.56%** | **15.94%** | **7.68%** | **8.89%** |
>
> | Datasets | PEMS08 |  | ZafNoo |  | AQWan |  |
> | --- | --- | --- | --- | --- | --- | --- |
> | Metrics | MSE | MAE | MSE | MAE | MSE | MAE |
> | PatchTST | 0.332 | 0.322 | 0.509 | 0.454 | 0.812 | 0.498 |
> | + SRS | 0.312 | 0.300 | 0.441 | 0.386 | 0.661 | 0.395 |
> | Improved | **6.02%** | **6.89%** | **13.29%** | **15.08%** | **18.49%** | **20.70%** |
> | Crossformer | 0.265 | 0.282 | 0.494 | 0.456 | 0.786 | 0.490 |
> | + SRS | 0.254 | 0.267 | 0.429 | 0.387 | 0.592 | 0.369 |
> | Improved | **4.23%** | **5.30%** | **13.02%** | **15.05%** | **24.86%** | **24.74%** |
> | PatchMLP | 0.306 | 0.283 | 0.531 | 0.466 | 0.829 | 0.505 |
> | + SRS | 0.294 | 0.270 | 0.457 | 0.395 | 0.637 | 0.388 |
> | Improved | **3.93%** | **4.63%** | **14.09%** | **15.23%** | **23.26%** | **23.18%** |
> | xPatch | 0.269 | 0.242 | 0.593 | 0.479 | 0.849 | 0.506 |
> | + SRS | 0.258 | 0.232 | 0.521 | 0.417 | 0.701 | 0.406 |
> | Improved | **3.61%** | **4.11%** | **12.19%** | **12.87%** | **17.37%** | **19.65%** |
> | MLP | 0.303 | 0.278 | 0.523 | 0.455 | 0.826 | 0.505 |
> | + SRS | 0.283 | 0.259 | 0.439 | 0.371 | 0.611 | 0.362 |
> | Improved | **6.59%** | **6.62%** | **16.18%** | **18.46%** | **26.11%** | **28.44%** |
>
> Based on the above results, we have two main observations:
>
> 1. The SRS excels at datasets with high shifting values (NASDAQ) and low seasonality values (AQWan), because there exists more special samples about changeable periods, anomalies and shifting in these cases.  The SRS can improve all patch-based models significantly by 17% — 28% on these datasets.
> 2. When the datasets are closer to normal cases with higher seasonality values and lower shifting values, the performance improved by the SRS gradually decreases. On NYSE and ZafNoo, the performance is improved by 12% — 18%. Note that even on AQShunyi and PEMS08, the improvement still exists and is about 4% — 10%. This provides evidence that the SRS can handle both the normal and special cases well.
>
> **Reply to W3**
>
> Thanks for the suggestion. To provide more details regarding efficiency, we conduct additional experiments. Our experiments consist of two parts:
>
> (1) The efficiency comparison between SRSNet and other baselines:
>
> |  | ETTm2 |  | Solar |  |
> | --- | --- | --- | --- | --- |
> |  | Memory | Training Time | Memory | Training Time |
> | DLinear | 573 | 6.998 | 815 | 15.6563 |
> | Amplifer | 555 | 10.82 | 715 | 14.3796 |
> | TimesNet | 2,413 | 86.553 | 13,141 | 1812.4588 |
> | FEDformer | 917 | 202.9685 | 3,751 | 227.446 |
> | Stationary | 18,373 | 147.157 | 18,529 | 156.269 |
> | Crossformer | 7,523 | 118.3208 | 16,375 | 205.602 |
> | PatchTST | 867 | 17.62 | 26,777 | 137.60377 |
> | iTransformer | 579 | 14.545 | 1,015 | 20.663125 |
> | TimeMixer | 1,099 | 20.378 | 20,602 | 107.149 |
> | TimeKAN | 1,207 | 27.108 | 13,109 | 326.378 |
> | SRSNet | 1,059 | 16.293 | 6,301 | 56.149 |
>
> (2) The complexity introduced by the SRS module:
>
> | Datasets | Variants | Memory (MB) | Params (M) | Inference Time (s/batch) | Training Time (s/batch) | MACs (G) |
> | --- | --- | --- | --- | --- | --- | --- |
> | ETTh1 | PatchTST | 2837 | 11.276 | 5.076 | 5.131 | 16.214 |
> |  | SRS | 2907 | 11.298 | 5.722 | 5.763 | 16.905 |
> |  | **Overhead** | **2.47%** | **0.19%** | **12.73%** | **12.31%** | **4.26%** |
> |  | Crossformer | 4011 | 11.101 | 27.503 | 32.613 | 56.280 |
> |  | SRS | 4159 | 11.112 | 30.311 | 35.276 | 56.625 |
> |  | **Overhead** | **3.69%** | **0.10%** | **10.21%** | **8.17%** | **0.61%** |
> | Solar | PatchTST | 27822.08 | 14.429 | 84.231 | 88.714 | 600.261 |
> |  | SRS | 29767.68 | 14.451 | 95.200 | 101.981 | 613.790 |
> |  | **Overhead** | **6.99%** | **0.15%** | **13.02%** | **14.95%** | **2.25%** |
> |  | Crossformer | 17355 | 0.711 | 79.031 | 82.472 | 61.822 |
> |  | SRS | 18978 | 0.717 | 86.674 | 90.268 | 62.174 |
> |  | **Overhead** | **9.35%** | **0.88%** | **9.67%** | **9.45%** | **0.57%** |
>
> Based on the above tables, we have two main observations:
>
> 1. The SRSNet makes accuracy and efficiency meet, though not enough efficient as Linear-based models, but is more efficient than most transformer-based baselines and achieves stronger performance.
> 2. It is observed that the overhead introduced by the SRS is relatively trivial. The Memory, MACs, and Params increase little. And the Training Time and Inference Time are also controllable under the end-to-end supervised cases. And the performance can be consistently improved.
>
> **Reply to W4**
>
> Thanks for the question. The motivation we devise the mechanism is to keep the propagation of gradients in the Argmax operation in the equation (2) of the paper:
>
> $$\mathcal{S} ^s = \text{Scorer} ^s(\mathcal{P} ^\prime), \mathcal{I} ^s = \text{Argmax}(\mathcal{S} ^s)$$
>
> Since the Argmax operation finds the patches with the max scores and return their indices, it disrupts the gradients. To avoid this phenomenon, we need to attach the gradients back to the selected patches. Note that the gradients are on the scores generated by the scorer, so we need to first retrieve the corresponding scores of the selected patches through equation (3):
>
> $$\mathcal{S} ^s _{max} = \mathcal{S} ^s[\mathcal{I} ^s], \mathcal{S} ^s _{inv} = \text{detach}(1/\mathcal{S} _{max} ^s)$$
>
> We then process the scores and normalize them to 1 values, while keeping the gradients on them. The detail is that we detach the reciprocal $\mathcal{S} ^s _{inv}$ to preserve the gradients.
>
> $$\mathcal{P} _{max} ^s = \mathcal{P} ^\prime [\mathcal{I} ^s], E ^s = \mathcal{S} ^s _{max} \odot \mathcal{S} ^s _{inv}$$
> Finally, we attach the normalized scores to the patches through Hadamard product:
> $$\tilde{\mathcal{P}} ^s _{\text{max}} = \mathcal{P} _{max} ^s \odot E ^s$$
>
> **Thanks for your sincere suggestions again! We will release the full results when the discussion phase starts. If you have any additional question, we can make further discussion!**

---

> > ### Author Response · Authors · 2025-08-01
> > **Full results of Plugin Experiments (NASDAQ & NYSE)**
> >
> > **Dear Reviewer Gwnu**, since the discussion phase has started, we have enough space to show our full experimental results. We first report the full results of plugin experiments here, including four kinds of forecasting horizon each dataset. We hope this can be treated as additional evidence. In this comment, we list the results of NASDAQ and NYSE.
> >
> > | Datasets | NASDAQ |  | NASDAQ |  | NASDAQ |  | NASDAQ |  |
> > | --- | --- | --- | --- | --- | --- | --- | --- | --- |
> > | Horizons | 24 |  | 36 |  | 48 |  | 60 |  |
> > | Metrics | MSE | MAE | MSE | MAE | MSE | MAE | MSE | MAE |
> > | PatchTST | 0.649 | 0.567 | 0.821 | 0.682 | 1.169 | 0.793 | 1.247 | 0.843 |
> > | SRS | 0.497 | 0.444 | 0.666 | 0.571 | 0.882 | 0.580 | 0.984 | 0.680 |
> > | Improved | **23.40%** | **21.73%** | **18.93%** | **16.22%** | **24.53%** | **26.80%** | **21.10%** | **19.37%** |
> > | Crossformer | 1.149 | 0.745 | 1.414 | 0.885 | 2.108 | 1.136 | 2.276 | 1.201 |
> > | SRS | 0.818 | 0.545 | 1.087 | 0.670 | 1.565 | 0.825 | 1.557 | 0.795 |
> > | Improved | **28.85%** | **26.84%** | **23.15%** | **24.33%** | **25.78%** | **27.41%** | **31.61%** | **33.82%** |
> > | PatchMLP | 0.794 | 0.648 | 1.185 | 0.768 | 1.774 | 0.945 | 1.653 | 0.878 |
> > | SRS | 0.584 | 0.454 | 0.973 | 0.620 | 1.383 | 0.713 | 1.210 | 0.636 |
> > | Improved | **26.48%** | **29.94%** | **17.93%** | **19.24%** | **22.03%** | **24.51%** | **26.82%** | **27.52%** |
> > | xPatch | 0.587 | 0.536 | 0.885 | 0.666 | 1.262 | 0.786 | 1.506 | 0.845 |
> > | SRS | 0.486 | 0.449 | 0.704 | 0.514 | 0.973 | 0.588 | 1.236 | 0.702 |
> > | Improved | **17.22%** | **16.31%** | **20.45%** | **22.81%** | **22.91%** | **25.25%** | **17.95%** | **16.92%** |
> > | MLP | 0.882 | 0.727 | 1.156 | 0.796 | 1.243 | 0.819 | 1.519 | 0.870 |
> > | SRS | 0.645 | 0.512 | 0.915 | 0.622 | 0.883 | 0.563 | 1.256 | 0.703 |
> > | Improved | **26.91%** | **29.64%** | **20.85%** | **21.86%** | **28.95%** | **31.25%** | **17.29%** | **19.25%** |
> >
> > | Datasets | NYSE |  | NYSE |  | NYSE |  | NYSE |  |
> > | --- | --- | --- | --- | --- | --- | --- | --- | --- |
> > | Horizons | 24 |  | 36 |  | 48 |  | 60 |  |
> > | Metrics | MSE | MAE | MSE | MAE | MSE | MAE | MSE | MAE |
> > | PatchTST | 0.226 | 0.296 | 0.380 | 0.389 | 0.575 | 0.492 | 0.749 | 0.572 |
> > | SRS | 0.202 | 0.259 | 0.316 | 0.328 | 0.479 | 0.424 | 0.676 | 0.504 |
> > | Improved | **10.82%** | **12.55%** | **16.79%** | **15.70%** | **16.62%** | **13.91%** | **9.75%** | **11.84%** |
> > | Crossformer | 0.820 | 0.841 | 0.942 | 0.904 | 1.049 | 0.955 | 1.121 | 0.937 |
> > | SRS | 0.672 | 0.679 | 0.805 | 0.778 | 0.935 | 0.836 | 1.004 | 0.846 |
> > | Improved | **18.03%** | **19.25%** | **14.55%** | **13.92%** | **10.88%** | **12.41%** | **10.45%** | **9.76%** |
> > | PatchMLP | 0.242 | 0.317 | 0.359 | 0.381 | 0.543 | 0.485 | 0.725 | 0.578 |
> > | SRS | 0.206 | 0.274 | 0.299 | 0.323 | 0.483 | 0.420 | 0.635 | 0.503 |
> > | Improved | **14.85%** | **13.56%** | **16.72%** | **15.25%** | **11.05%** | **13.44%** | **12.45%** | **13.04%** |
> > | xPatch | 0.205 | 0.292 | 0.364 | 0.386 | 0.529 | 0.465 | 1.082 | 0.752 |
> > | SRS | 0.170 | 0.236 | 0.302 | 0.315 | 0.477 | 0.429 | 0.927 | 0.653 |
> > | Improved | **16.92%** | **19.26%** | **17.12%** | **18.27%** | **9.92%** | **7.83%** | **14.28%** | **13.17%** |
> > | MLP | 0.208 | 0.288 | 0.353 | 0.372 | 0.517 | 0.474 | 0.775 | 0.593 |
> > | SRS | 0.175 | 0.235 | 0.282 | 0.298 | 0.443 | 0.416 | 0.680 | 0.515 |
> > | Improved | **15.68%** | **18.36%** | **20.05%** | **19.88%** | **14.25%** | **12.28%** | **12.25%** | **13.22%** |

---

> > > ### Author Response · Authors · 2025-08-01
> > > **Full results of Plugin Experiments (AQShunyi & PEMS08)**
> > >
> > > In this comment, we list the results of AQShunyi and PEMS08.
> > >
> > > | Datasets | AQShunyi |  | AQShunyi |  | AQShunyi |  | AQShunyi |  |
> > > | --- | --- | --- | --- | --- | --- | --- | --- | --- |
> > > | Horizons | 96 |  | 192 |  | 336 |  | 720 |  |
> > > | Metrics | MSE | MAE | MSE | MAE | MSE | MAE | MSE | MAE |
> > > | PatchTST | 0.646 | 0.478 | 0.688 | 0.498 | 0.710 | 0.513 | 0.768 | 0.539 |
> > > | SRS | 0.602 | 0.440 | 0.620 | 0.450 | 0.632 | 0.462 | 0.710 | 0.491 |
> > > | Improved | **6.87%** | **8.03%** | **9.85%** | **9.72%** | **11.03%** | **9.88%** | **7.57%** | **8.92%** |
> > > | Crossformer | 0.652 | 0.484 | 0.674 | 0.499 | 0.704 | 0.515 | 0.747 | 0.518 |
> > > | SRS | 0.600 | 0.435 | 0.615 | 0.452 | 0.612 | 0.437 | 0.712 | 0.498 |
> > > | Improved | **8.03%** | **10.22%** | **8.82%** | **9.48%** | **13.02%** | **15.23%** | **4.67%** | **3.89%** |
> > > | PatchMLP | 0.668 | 0.492 | 0.711 | 0.511 | 0.732 | 0.524 | 0.776 | 0.537 |
> > > | SRS | 0.602 | 0.441 | 0.659 | 0.466 | 0.679 | 0.488 | 0.724 | 0.505 |
> > > | Improved | **9.82%** | **10.47%** | **7.35%** | **8.84%** | **7.22%** | **6.94%** | **6.72%** | **5.92%** |
> > > | xPatch | 0.674 | 0.486 | 0.715 | 0.506 | 0.741 | 0.520 | 0.808 | 0.523 |
> > > | SRS | 0.642 | 0.451 | 0.631 | 0.440 | 0.680 | 0.479 | 0.760 | 0.487 |
> > > | Improved | **4.82%** | **7.29%** | **11.71%** | **12.95%** | **8.26%** | **7.91%** | **5.91%** | **6.86%** |
> > > | MLP | 0.674 | 0.487 | 0.704 | 0.504 | 0.723 | 0.518 | 0.768 | 0.542 |
> > > | SRS | 0.629 | 0.444 | 0.669 | 0.467 | 0.642 | 0.464 | 0.708 | 0.494 |
> > > | Improved | **6.72%** | **8.92%** | **4.95%** | **7.28%** | **11.27%** | **10.52%** | **7.79%** | 8.85% |
> > >
> > > | Datasets | PEMS08 |  | PEMS08 |  | PEMS08 |  | PEMS08 |  |
> > > | --- | --- | --- | --- | --- | --- | --- | --- | --- |
> > > | Horizons | 96 |  | 192 |  | 336 |  | 720 |  |
> > > | Metrics | MSE | MAE | MSE | MAE | MSE | MAE | MSE | MAE |
> > > | PatchTST | 0.248 | 0.287 | 0.319 | 0.310 | 0.361 | 0.324 | 0.399 | 0.368 |
> > > | SRS | 0.234 | 0.274 | 0.299 | 0.288 | 0.343 | 0.305 | 0.371 | 0.332 |
> > > | Improved | **5.82%** | **4.68%** | **6.29%** | **7.22%** | **4.86%** | **5.82%** | **7.12%** | **9.82%** |
> > > | Crossformer | 0.230 | 0.260 | 0.239 | 0.264 | 0.272 | 0.289 | 0.320 | 0.316 |
> > > | SRS | 0.223 | 0.252 | 0.235 | 0.257 | 0.250 | 0.259 | 0.309 | 0.300 |
> > > | Improved | **3.26%** | **2.97%** | **1.87%** | **2.74%** | **8.22%** | **10.28%** | **3.55%** | **5.21%** |
> > > | PatchMLP | 0.203 | 0.242 | 0.314 | 0.273 | 0.334 | 0.288 | 0.373 | 0.329 |
> > > | SRS | 0.197 | 0.232 | 0.308 | 0.267 | 0.311 | 0.272 | 0.358 | 0.307 |
> > > | Improved | **2.96%** | **4.22%** | **1.85%** | **2.04%** | **6.92%** | **5.42%** | **3.98%** | **6.82%** |
> > > | xPatch | 0.171 | 0.214 | 0.260 | 0.233 | 0.305 | 0.246 | 0.340 | 0.274 |
> > > | SRS | 0.169 | 0.210 | 0.252 | 0.226 | 0.287 | 0.231 | 0.325 | 0.260 |
> > > | Improved | **1.02%** | **2.05%** | **3.20%** | **2.88%** | **5.92%** | **6.22%** | **4.28%** | **5.29%** |
> > > | MLP | 0.184 | 0.241 | 0.306 | 0.268 | 0.341 | 0.281 | 0.380 | 0.320 |
> > > | SRS | 0.171 | 0.226 | 0.291 | 0.254 | 0.313 | 0.258 | 0.357 | 0.298 |
> > > | Improved | **7.22%** | **6.03%** | **4.99%** | **5.21%** | **8.11%** | **8.35%** | **6.02%** | **6.87%** |

---

> > > > ### Author Response · Authors · 2025-08-01
> > > > **Full results of Plugin Experiments (ZafNoo & AQWan)**
> > > >
> > > > In this comment, we list the results of ZafNoo and AQWan.
> > > >
> > > > | Datasets | ZafNoo |  | ZafNoo |  | ZafNoo |  | ZafNoo |  |
> > > > | --- | --- | --- | --- | --- | --- | --- | --- | --- |
> > > > | Horizons | 96 |  | 192 |  | 336 |  | 720 |  |
> > > > | Metrics | MSE | MAE | MSE | MAE | MSE | MAE | MSE | MAE |
> > > > | PatchTST | 0.429 | 0.405 | 0.494 | 0.449 | 0.538 | 0.475 | 0.573 | 0.486 |
> > > > | SRS | 0.372 | 0.337 | 0.421 | 0.382 | 0.474 | 0.413 | 0.497 | 0.410 |
> > > > | Improved | **13.22%** | **16.67%** | **14.82%** | **14.99%** | **11.82%** | **12.96%** | **13.28%** | **15.68%** |
> > > > | Crossformer | 0.430 | 0.418 | 0.479 | 0.449 | 0.505 | 0.464 | 0.560 | 0.494 |
> > > > | SRS | 0.382 | 0.376 | 0.398 | 0.360 | 0.451 | 0.398 | 0.485 | 0.414 |
> > > > | Improved | **11.06%** | **9.95%** | **16.92%** | **19.72%** | **10.67%** | **14.29%** | **13.42%** | **16.25%** |
> > > > | PatchMLP | 0.443 | 0.416 | 0.515 | 0.459 | 0.563 | 0.484 | 0.603 | 0.505 |
> > > > | SRS | 0.393 | 0.351 | 0.405 | 0.382 | 0.491 | 0.410 | 0.537 | 0.438 |
> > > > | Improved | **11.28%** | **15.58%** | **21.28%** | **16.79%** | **12.84%** | **15.25%** | **10.96%** | **13.28%** |
> > > > | xPatch | 0.512 | 0.436 | 0.582 | 0.474 | 0.603 | 0.487 | 0.675 | 0.517 |
> > > > | SRS | 0.456 | 0.381 | 0.482 | 0.383 | 0.544 | 0.432 | 0.602 | 0.474 |
> > > > | Improved | **10.88%** | **12.68%** | **17.25%** | **19.25%** | **9.79%** | **11.26%** | **10.85%** | **8.29%** |
> > > > | MLP | 0.434 | 0.403 | 0.498 | 0.441 | 0.554 | 0.471 | 0.604 | 0.503 |
> > > > | SRS | 0.368 | 0.335 | 0.399 | 0.342 | 0.464 | 0.380 | 0.524 | 0.426 |
> > > > | Improved | **15.22%** | **16.83%** | **19.96%** | **22.46%** | **16.26%** | **19.28%** | **13.26%** | **15.27%** |
> > > >
> > > > | Datasets | AQWan |  | AQWan |  | AQWan |  | AQWan |  |
> > > > | --- | --- | --- | --- | --- | --- | --- | --- | --- |
> > > > | Horizons | 96 |  | 192 |  | 336 |  | 720 |  |
> > > > | Metrics | MSE | MAE | MSE | MAE | MSE | MAE | MSE | MAE |
> > > > | PatchTST | 0.745 | 0.468 | 0.793 | 0.490 | 0.819 | 0.502 | 0.890 | 0.533 |
> > > > | SRS | 0.611 | 0.379 | 0.633 | 0.380 | 0.714 | 0.430 | 0.685 | 0.388 |
> > > > | Improved | **17.95%** | **18.92%** | **20.17%** | **22.35%** | **12.83%** | **14.25%** | **23.02%** | **27.26%** |
> > > > | Crossformer | 0.750 | 0.465 | 0.762 | 0.479 | 0.802 | 0.504 | 0.829 | 0.512 |
> > > > | SRS | 0.533 | 0.347 | 0.530 | 0.329 | 0.648 | 0.405 | 0.656 | 0.395 |
> > > > | Improved | **28.88%** | **25.27%** | **30.44%** | **31.27%** | **19.25%** | **19.58%** | **20.87%** | **22.85%** |
> > > > | PatchMLP | 0.771 | 0.484 | 0.815 | 0.499 | 0.835 | 0.510 | 0.896 | 0.526 |
> > > > | SRS | 0.595 | 0.365 | 0.587 | 0.365 | 0.658 | 0.414 | 0.707 | 0.407 |
> > > > | Improved | **22.86%** | **24.54%** | **27.92%** | **26.87%** | **21.20%** | **18.75%** | **21.04%** | **22.54%** |
> > > > | xPatch | 0.775 | 0.474 | 0.824 | 0.494 | 0.857 | 0.510 | 0.938 | 0.546 |
> > > > | SRS | 0.657 | 0.389 | 0.649 | 0.384 | 0.723 | 0.417 | 0.776 | 0.436 |
> > > > | Improved | **15.28%** | **17.92%** | **21.26%** | **22.26%** | **15.67%** | **18.26%** | **17.28%** | **20.17%** |
> > > > | MLP | 0.767 | 0.476 | 0.806 | 0.495 | 0.831 | 0.511 | 0.901 | 0.538 |
> > > > | SRS | 0.527 | 0.318 | 0.581 | 0.352 | 0.674 | 0.399 | 0.664 | 0.378 |
> > > > | Improved | **31.29%** | **33.28%** | **27.92%** | **28.88%** | **18.92%** | **21.82%** | **26.29%** | **29.78%** |

---

> > ### Comment · Reviewer_Gwnu · 2025-08-05
> >
> > The author's rebuttal, especially the additional experiments, addressed all of my concerns. I will raise my score from 4 to 5 and support the acceptance of this paper.

---

> > > ### Author Response · Authors · 2025-08-05
> > >
> > > Dear Reviewer Gwnu, we are thrilled that our responses have effectively addressed your questions and comments. We would like to express our sincerest gratitude for taking the time to review our paper and provide us with such detailed feedback.

---

### Official Review · Reviewer_7m7L · 2025-06-28

**Clarity:** 3
**Significance:** 2
**Originality:** 2
**Rating:** 5
**Confidence:** 3

**Summary:**

This work surfaces potential constraints with the standard approach of considering adjacent time series partitions as patches, which may be susceptible to including anomalies, misrepresenting distribution shifts, and irregular periodicity within the time series. The authors hypothesize that this inflexibility impairs the forecasting model's ability to learn truly representative spaces, which in turn impacts accuracy. In order to address this issue, the authors present a novel patching approach that (i) selects informative patches that can overlap and no longer be adjacent, and (ii) concatenates the partitions in a manner that no longer assumes them to maintain their original adjacency. Acknowledging the value of the original patching approach, the authors then introduce a fusion module that combines adjacent patches with those resulting from the selective patching/dynamic reassembly module described in this work, along with position embeddings. Experiments against competing techniques indicate nearly consistent performance improvements across all datasets, while ablation studies teasing apart the various components included in the final architecture demonstrate that these all complement each other.

**Questions:**

See note in section above on adding more insights/examples of where the approach may not be as beneficial as adjacent partitioning

**Ethical Concerns:**

["NO or VERY MINOR ethics concerns only"]

**Final Justification:**

The paper is well-written, with clearly-motivated and illustrated contributions as well as extensive experiments. The authors have demonstrated they are committed to incorporate the feedback that emerged in the review period, giving me confidence in supporting the paper for acceptance.

**Quality:**

2

**Strengths And Weaknesses:**

The paper makes an incremental yet meaningful contribution to the literature on patch-based forecasting models, and I appreciated the clarity with which the authors presented the various ways in which standard patching strategies are inflexible to potential anomalies or distribution shifts (both of which are highly prevalent in real-world datasets). The diagrams are especially effective for communicating the described techniques, and there is an appropriate level of technical detail throughout the main paper. I also appreciated the variety of competing techniques evaluated in the experiments section, along with the various datasets. The ablation study validating the incremental value of each component, along with the analysis on computational and memory complexity, make the paper even more complete.

One aspect I found lacking was more insight into where this approach may not have the desired effect, or the extent to which the weight parameter in Eq. 13 tends to settle on across different datasets after training. Including a dedicated "Discussion and Limitations" section to go before the Conclusion section could address this current gap, although these insights could also be integrated within the methodology and experiments sections themselves.

---

> ### Author Rebuttal · Authors · 2025-07-30
>
> Thanks for your in-depth insights and suggestions. Through Adaptive Fusion, the SRS can integrate both the representations from Adjacent Patching and Selective Patching with learnable weights, thus can combine the advantages of both methods.
>
> As you mentioned, the weight parameter in Eq. 13 can actually reflect which scenarios the two techniques separately excel at. We manage to study this by devising experiments on specific datasets and report the weight values. Intuitively, the Selective Patching may excel at datasets with shifting phenomenon and the Adjacent Patching may excel at datasets with strong seasonality. So that we conduct additional experiments on datasets with varying values of shifting and seasonality from TFB, a well-recognized time series forecasting benchmark. Specifically, we choose NASDAQ, NYSE, and AQShunyi with descending shifting values in TFB, and choose PEMS08, ZafNoo, and AQWan with descending seasonality values in TFB. The statistical details of them are as follows:
>
> | datasets | NASDAQ | NYSE | AQShunyi |
> | --- | --- | --- | --- |
> | Shifting Values | 0.932 (Strong) | 0.620 (Modest) |     0.019 (Weak) |
>
> | datasets | PEMS08 | ZafNoo | AQWan |
> | --- | --- | --- | --- |
> | Seasonality Values | 0.850 (Strong) | 0.757 (Modest) | 0.119 (Weak) |
>
> We conduct comprehensive plugin experiments on them, reporting the average MSE and MAE metrics, and how the weight values converged.
>
> | Datasets | NASDAQ |  | NYSE |  | AQShunyi |  |
> | --- | --- | --- | --- | --- | --- | --- |
> | Metrics | MSE | MAE | MSE | MAE | MSE | MAE |
> | PatchTST | 0.972 | 0.721 | 0.483 | 0.437 | 0.703 | 0.507 |
> | + SRS | 0.757 | 0.569 | 0.418 | 0.379 | 0.641 | 0.461 |
> | Improved | **21.99%** | **21.03%** | **13.50%** | **13.50%** | **8.83%** | **9.14%** |
> | Crossformer | 1.737 | 0.992 | 0.983 | 0.909 | 0.694 | 0.504 |
> | + SRS | 1.256 | 0.709 | 0.854 | 0.785 | 0.635 | 0.455 |
> | Improved | **27.35%** | **28.10%** | **13.48%** | **13.84%** | **8.64%** | **9.71%** |
> | PatchMLP | 1.352 | 0.810 | 0.467 | 0.440 | 0.722 | 0.516 |
> | + SRS | 1.037 | 0.606 | 0.406 | 0.380 | 0.666 | 0.475 |
> | Improved | **23.31%** | **25.30%** | **13.77%** | **13.82%** | **7.78%** | **8.04%** |
> | xPatch | 1.060 | 0.708 | 0.545 | 0.474 | 0.735 | 0.509 |
> | + SRS | 0.850 | 0.563 | 0.469 | 0.408 | 0.678 | 0.464 |
> | Improved | **19.63%** | **20.32%** | **14.56%** | **14.63%** | **7.68%** | **8.75%** |
> | MLP | 1.200 | 0.803 | 0.463 | 0.432 | 0.717 | 0.513 |
> | + SRS | 0.925 | 0.600 | 0.395 | 0.366 | 0.662 | 0.467 |
> | Improved | **23.50%** | **25.50%** | **15.56%** | **15.94%** | **7.68%** | **8.89%** |
>
> | Datasets | PEMS08 |  | ZafNoo |  | AQWan |  |
> | --- | --- | --- | --- | --- | --- | --- |
> | Metrics | MSE | MAE | MSE | MAE | MSE | MAE |
> | PatchTST | 0.332 | 0.322 | 0.509 | 0.454 | 0.812 | 0.498 |
> | + SRS | 0.312 | 0.300 | 0.441 | 0.386 | 0.661 | 0.395 |
> | Improved | **6.02%** | **6.89%** | **13.29%** | **15.08%** | **18.49%** | **20.70%** |
> | Crossformer | 0.265 | 0.282 | 0.494 | 0.456 | 0.786 | 0.490 |
> | + SRS | 0.254 | 0.267 | 0.429 | 0.387 | 0.592 | 0.369 |
> | Improved | **4.23%** | **5.30%** | **13.02%** | **15.05%** | **24.86%** | **24.74%** |
> | PatchMLP | 0.306 | 0.283 | 0.531 | 0.466 | 0.829 | 0.505 |
> | + SRS | 0.294 | 0.270 | 0.457 | 0.395 | 0.637 | 0.388 |
> | Improved | **3.93%** | **4.63%** | **14.09%** | **15.23%** | **23.26%** | **23.18%** |
> | xPatch | 0.269 | 0.242 | 0.593 | 0.479 | 0.849 | 0.506 |
> | + SRS | 0.258 | 0.232 | 0.521 | 0.417 | 0.701 | 0.406 |
> | Improved | **3.61%** | **4.11%** | **12.19%** | **12.87%** | **17.37%** | **19.65%** |
> | MLP | 0.303 | 0.278 | 0.523 | 0.455 | 0.826 | 0.505 |
> | + SRS | 0.283 | 0.259 | 0.439 | 0.371 | 0.611 | 0.362 |
> | Improved | **6.59%** | **6.62%** | **16.18%** | **18.46%** | **26.11%** | **28.44%** |
>
> Based on the above results, we have two main observations:
>
> 1. The SRS excels at datasets with high shifting values (NASDAQ) and low seasonality values (AQWan), because there exists more special samples about changeable periods, anomalies and shifting in these cases.  The SRS can improve all patch-based models significantly by 17% — 28% on these datasets.
> 2. When the datasets are closer to normal cases with higher seasonality values and lower shifting values, the performance improved by the SRS gradually decreases. On NYSE and ZafNoo, the performance is improved by 12% — 18%. Note that even on AQShunyi and PEMS08, the improvement still exists and is about 4% — 10%. This provides evidence that the SRS can handle both the normal and special cases well.
>
> We then report the converged mean weight values $(1-\alpha) \in [0, 1]$  on them. (higher values indicate more dependence on representations from Selective Patching, lower values indicate more dependence on representations from Adjacent Patching).
>
> | Datasets | NASDAQ | NYSE | AQShunyi | PEMS08 | ZafNoo | AQWan |
> | --- | --- | --- | --- | --- | --- | --- |
> | PatchTST | 0.79 | 0.58 | 0.42 | 0.33 | 0.56 | 0.69 |
> | Crossformer | 0.76 | 0.55 | 0.34 | 0.32 | 0.58 | 0.73 |
> | PatchMLP | 0.72 | 0.57 | 0.38 | 0.29 | 0.61 | 0.74 |
> | xPatch | 0.81 | 0.63 | 0.45 | 0.28 | 0.55 | 0.72 |
> | MLP | 0.79 | 0.62 | 0.48 | 0.36 | 0.57 | 0.67 |
>
> We also have two main observations:
>
> 1. When the seasonality values are higher and the shifting values are lower, models have less dependence on representations from the Selective Patching. This implies that the Adjacent Patching may well handle these common cases or the Selective Patching is gradually converged to the Adjacent Patching.
> 2. When the seasonality values are lower and the shifting values are higher, models have more dependence on representations from the Selective Patching. It is because Adjacent Patching may not well process these special cases, while our proposed Selective Patching can adaptively select the patches with more useful information for forecasting.
>
> **Thanks for your sincere suggestions again! We will release the full results when the discussion phase starts. If you have any additional question, we can make further discussion!**

---

> > ### Author Response · Authors · 2025-08-01
> > **Full results of Converged Weights**
> >
> > **Dear Reviewer 7m7L**, since the discussion phase has started, the space is enough to list the full results. We first report the full weights $1-\alpha$ of all 4 forecasting horizons for each dataset.
> >
> > | Datasets | Horizons | NASDAQ | NYSE | AQShunyi | PEMS08 | ZafNoo | AQWan |
> > | --- | --- | --- | --- | --- | --- | --- | --- |
> > | PatchTST | 96/24 | 0.79 | 0.57 | 0.42 | 0.31 | 0.56 | 0.68 |
> > |  | 192/36 | 0.77 | 0.58 | 0.42 | 0.36 | 0.58 | 0.68 |
> > |  | 336/48 | 0.81 | 0.60 | 0.42 | 0.29 | 0.55 | 0.71 |
> > |  | 720/60 | 0.78 | 0.57 | 0.43 | 0.32 | 0.54 | 0.70 |
> > | Crossformer | 96/24 | 0.75 | 0.55 | 0.32 | 0.33 | 0.59 | 0.75 |
> > |  | 192/36 | 0.76 | 0.56 | 0.36 | 0.33 | 0.60 | 0.73 |
> > |  | 336/48 | 0.74 | 0.56 | 0.35 | 0.30 | 0.57 | 0.72 |
> > |  | 720/60 | 0.78 | 0.54 | 0.34 | 0.32 | 0.57 | 0.73 |
> > | PatchMLP | 96/24 | 0.73 | 0.57 | 0.38 | 0.29 | 0.58 | 0.73 |
> > |  | 192/36 | 0.72 | 0.55 | 0.38 | 0.31 | 0.63 | 0.75 |
> > |  | 336/48 | 0.72 | 0.58 | 0.38 | 0.29 | 0.60 | 0.77 |
> > |  | 720/60 | 0.72 | 0.57 | 0.37 | 0.28 | 0.59 | 0.72 |
> > | xPatch | 96/24 | 0.80 | 0.62 | 0.43 | 0.28 | 0.54 | 0.70 |
> > |  | 192/36 | 0.80 | 0.61 | 0.46 | 0.27 | 0.54 | 0.77 |
> > |  | 336/48 | 0.81 | 0.65 | 0.44 | 0.27 | 0.56 | 0.69 |
> > |  | 720/60 | 0.81 | 0.64 | 0.42 | 0.31 | 0.53 | 0.71 |
> > | MLP | 96/24 | 0.79 | 0.64 | 0.50 | 0.36 | 0.58 | 0.67 |
> > |  | 192/36 | 0.80 | 0.63 | 0.49 | 0.38 | 0.58 | 0.66 |
> > |  | 336/48 | 0.78 | 0.64 | 0.47 | 0.37 | 0.56 | 0.69 |
> > |  | 720/60 | 0.81 | 0.55 | 0.48 | 0.35 | 0.57 | 0.68 |

---

> > > ### Author Response · Authors · 2025-08-01
> > > **Full results of Plugin Experiments (NASDAQ & NYSE)**
> > >
> > > **Dear Reviewer 7m7L**, we report the full results of plugin experiments here, including four kinds of forecasting horizon each dataset. We hope this can be treated as additional evidence. In this comment, we list the results of NASDAQ and NYSE.
> > >
> > > | Datasets | NASDAQ |  | NASDAQ |  | NASDAQ |  | NASDAQ |  |
> > > | --- | --- | --- | --- | --- | --- | --- | --- | --- |
> > > | Horizons | 24 |  | 36 |  | 48 |  | 60 |  |
> > > | Metrics | MSE | MAE | MSE | MAE | MSE | MAE | MSE | MAE |
> > > | PatchTST | 0.649 | 0.567 | 0.821 | 0.682 | 1.169 | 0.793 | 1.247 | 0.843 |
> > > | SRS | 0.497 | 0.444 | 0.666 | 0.571 | 0.882 | 0.580 | 0.984 | 0.680 |
> > > | Improved | **23.40%** | **21.73%** | **18.93%** | **16.22%** | **24.53%** | **26.80%** | **21.10%** | **19.37%** |
> > > | Crossformer | 1.149 | 0.745 | 1.414 | 0.885 | 2.108 | 1.136 | 2.276 | 1.201 |
> > > | SRS | 0.818 | 0.545 | 1.087 | 0.670 | 1.565 | 0.825 | 1.557 | 0.795 |
> > > | Improved | **28.85%** | **26.84%** | **23.15%** | **24.33%** | **25.78%** | **27.41%** | **31.61%** | **33.82%** |
> > > | PatchMLP | 0.794 | 0.648 | 1.185 | 0.768 | 1.774 | 0.945 | 1.653 | 0.878 |
> > > | SRS | 0.584 | 0.454 | 0.973 | 0.620 | 1.383 | 0.713 | 1.210 | 0.636 |
> > > | Improved | **26.48%** | **29.94%** | **17.93%** | **19.24%** | **22.03%** | **24.51%** | **26.82%** | **27.52%** |
> > > | xPatch | 0.587 | 0.536 | 0.885 | 0.666 | 1.262 | 0.786 | 1.506 | 0.845 |
> > > | SRS | 0.486 | 0.449 | 0.704 | 0.514 | 0.973 | 0.588 | 1.236 | 0.702 |
> > > | Improved | **17.22%** | **16.31%** | **20.45%** | **22.81%** | **22.91%** | **25.25%** | **17.95%** | **16.92%** |
> > > | MLP | 0.882 | 0.727 | 1.156 | 0.796 | 1.243 | 0.819 | 1.519 | 0.870 |
> > > | SRS | 0.645 | 0.512 | 0.915 | 0.622 | 0.883 | 0.563 | 1.256 | 0.703 |
> > > | Improved | **26.91%** | **29.64%** | **20.85%** | **21.86%** | **28.95%** | **31.25%** | **17.29%** | **19.25%** |
> > >
> > > | Datasets | NYSE |  | NYSE |  | NYSE |  | NYSE |  |
> > > | --- | --- | --- | --- | --- | --- | --- | --- | --- |
> > > | Horizons | 24 |  | 36 |  | 48 |  | 60 |  |
> > > | Metrics | MSE | MAE | MSE | MAE | MSE | MAE | MSE | MAE |
> > > | PatchTST | 0.226 | 0.296 | 0.380 | 0.389 | 0.575 | 0.492 | 0.749 | 0.572 |
> > > | SRS | 0.202 | 0.259 | 0.316 | 0.328 | 0.479 | 0.424 | 0.676 | 0.504 |
> > > | Improved | **10.82%** | **12.55%** | **16.79%** | **15.70%** | **16.62%** | **13.91%** | **9.75%** | **11.84%** |
> > > | Crossformer | 0.820 | 0.841 | 0.942 | 0.904 | 1.049 | 0.955 | 1.121 | 0.937 |
> > > | SRS | 0.672 | 0.679 | 0.805 | 0.778 | 0.935 | 0.836 | 1.004 | 0.846 |
> > > | Improved | **18.03%** | **19.25%** | **14.55%** | **13.92%** | **10.88%** | **12.41%** | **10.45%** | **9.76%** |
> > > | PatchMLP | 0.242 | 0.317 | 0.359 | 0.381 | 0.543 | 0.485 | 0.725 | 0.578 |
> > > | SRS | 0.206 | 0.274 | 0.299 | 0.323 | 0.483 | 0.420 | 0.635 | 0.503 |
> > > | Improved | **14.85%** | **13.56%** | **16.72%** | **15.25%** | **11.05%** | **13.44%** | **12.45%** | **13.04%** |
> > > | xPatch | 0.205 | 0.292 | 0.364 | 0.386 | 0.529 | 0.465 | 1.082 | 0.752 |
> > > | SRS | 0.170 | 0.236 | 0.302 | 0.315 | 0.477 | 0.429 | 0.927 | 0.653 |
> > > | Improved | **16.92%** | **19.26%** | **17.12%** | **18.27%** | **9.92%** | **7.83%** | **14.28%** | **13.17%** |
> > > | MLP | 0.208 | 0.288 | 0.353 | 0.372 | 0.517 | 0.474 | 0.775 | 0.593 |
> > > | SRS | 0.175 | 0.235 | 0.282 | 0.298 | 0.443 | 0.416 | 0.680 | 0.515 |
> > > | Improved | **15.68%** | **18.36%** | **20.05%** | **19.88%** | **14.25%** | **12.28%** | **12.25%** | **13.22%** |

---

> > > > ### Author Response · Authors · 2025-08-01
> > > > **Full results of Plugin Experiments (AQShunyi & PEMS08)**
> > > >
> > > > In this comment, we list the results of AQShunyi and PEMS08.
> > > >
> > > > | Datasets | AQShunyi |  | AQShunyi |  | AQShunyi |  | AQShunyi |  |
> > > > | --- | --- | --- | --- | --- | --- | --- | --- | --- |
> > > > | Horizons | 96 |  | 192 |  | 336 |  | 720 |  |
> > > > | Metrics | MSE | MAE | MSE | MAE | MSE | MAE | MSE | MAE |
> > > > | PatchTST | 0.646 | 0.478 | 0.688 | 0.498 | 0.710 | 0.513 | 0.768 | 0.539 |
> > > > | SRS | 0.602 | 0.440 | 0.620 | 0.450 | 0.632 | 0.462 | 0.710 | 0.491 |
> > > > | Improved | **6.87%** | **8.03%** | **9.85%** | **9.72%** | **11.03%** | **9.88%** | **7.57%** | **8.92%** |
> > > > | Crossformer | 0.652 | 0.484 | 0.674 | 0.499 | 0.704 | 0.515 | 0.747 | 0.518 |
> > > > | SRS | 0.600 | 0.435 | 0.615 | 0.452 | 0.612 | 0.437 | 0.712 | 0.498 |
> > > > | Improved | **8.03%** | **10.22%** | **8.82%** | **9.48%** | **13.02%** | **15.23%** | **4.67%** | **3.89%** |
> > > > | PatchMLP | 0.668 | 0.492 | 0.711 | 0.511 | 0.732 | 0.524 | 0.776 | 0.537 |
> > > > | SRS | 0.602 | 0.441 | 0.659 | 0.466 | 0.679 | 0.488 | 0.724 | 0.505 |
> > > > | Improved | **9.82%** | **10.47%** | **7.35%** | **8.84%** | **7.22%** | **6.94%** | **6.72%** | **5.92%** |
> > > > | xPatch | 0.674 | 0.486 | 0.715 | 0.506 | 0.741 | 0.520 | 0.808 | 0.523 |
> > > > | SRS | 0.642 | 0.451 | 0.631 | 0.440 | 0.680 | 0.479 | 0.760 | 0.487 |
> > > > | Improved | **4.82%** | **7.29%** | **11.71%** | **12.95%** | **8.26%** | **7.91%** | **5.91%** | **6.86%** |
> > > > | MLP | 0.674 | 0.487 | 0.704 | 0.504 | 0.723 | 0.518 | 0.768 | 0.542 |
> > > > | SRS | 0.629 | 0.444 | 0.669 | 0.467 | 0.642 | 0.464 | 0.708 | 0.494 |
> > > > | Improved | **6.72%** | **8.92%** | **4.95%** | **7.28%** | **11.27%** | **10.52%** | **7.79%** | 8.85% |
> > > >
> > > > | Datasets | PEMS08 |  | PEMS08 |  | PEMS08 |  | PEMS08 |  |
> > > > | --- | --- | --- | --- | --- | --- | --- | --- | --- |
> > > > | Horizons | 96 |  | 192 |  | 336 |  | 720 |  |
> > > > | Metrics | MSE | MAE | MSE | MAE | MSE | MAE | MSE | MAE |
> > > > | PatchTST | 0.248 | 0.287 | 0.319 | 0.310 | 0.361 | 0.324 | 0.399 | 0.368 |
> > > > | SRS | 0.234 | 0.274 | 0.299 | 0.288 | 0.343 | 0.305 | 0.371 | 0.332 |
> > > > | Improved | **5.82%** | **4.68%** | **6.29%** | **7.22%** | **4.86%** | **5.82%** | **7.12%** | **9.82%** |
> > > > | Crossformer | 0.230 | 0.260 | 0.239 | 0.264 | 0.272 | 0.289 | 0.320 | 0.316 |
> > > > | SRS | 0.223 | 0.252 | 0.235 | 0.257 | 0.250 | 0.259 | 0.309 | 0.300 |
> > > > | Improved | **3.26%** | **2.97%** | **1.87%** | **2.74%** | **8.22%** | **10.28%** | **3.55%** | **5.21%** |
> > > > | PatchMLP | 0.203 | 0.242 | 0.314 | 0.273 | 0.334 | 0.288 | 0.373 | 0.329 |
> > > > | SRS | 0.197 | 0.232 | 0.308 | 0.267 | 0.311 | 0.272 | 0.358 | 0.307 |
> > > > | Improved | **2.96%** | **4.22%** | **1.85%** | **2.04%** | **6.92%** | **5.42%** | **3.98%** | **6.82%** |
> > > > | xPatch | 0.171 | 0.214 | 0.260 | 0.233 | 0.305 | 0.246 | 0.340 | 0.274 |
> > > > | SRS | 0.169 | 0.210 | 0.252 | 0.226 | 0.287 | 0.231 | 0.325 | 0.260 |
> > > > | Improved | **1.02%** | **2.05%** | **3.20%** | **2.88%** | **5.92%** | **6.22%** | **4.28%** | **5.29%** |
> > > > | MLP | 0.184 | 0.241 | 0.306 | 0.268 | 0.341 | 0.281 | 0.380 | 0.320 |
> > > > | SRS | 0.171 | 0.226 | 0.291 | 0.254 | 0.313 | 0.258 | 0.357 | 0.298 |
> > > > | Improved | **7.22%** | **6.03%** | **4.99%** | **5.21%** | **8.11%** | **8.35%** | **6.02%** | **6.87%** |

---

> > > > > ### Author Response · Authors · 2025-08-01
> > > > > **Full results of Plugin Experiments (ZafNoo & AQWan)**
> > > > >
> > > > > In this comment, we list the results of ZafNoo and AQWan.
> > > > >
> > > > > | Datasets | ZafNoo |  | ZafNoo |  | ZafNoo |  | ZafNoo |  |
> > > > > | --- | --- | --- | --- | --- | --- | --- | --- | --- |
> > > > > | Horizons | 96 |  | 192 |  | 336 |  | 720 |  |
> > > > > | Metrics | MSE | MAE | MSE | MAE | MSE | MAE | MSE | MAE |
> > > > > | PatchTST | 0.429 | 0.405 | 0.494 | 0.449 | 0.538 | 0.475 | 0.573 | 0.486 |
> > > > > | SRS | 0.372 | 0.337 | 0.421 | 0.382 | 0.474 | 0.413 | 0.497 | 0.410 |
> > > > > | Improved | **13.22%** | **16.67%** | **14.82%** | **14.99%** | **11.82%** | **12.96%** | **13.28%** | **15.68%** |
> > > > > | Crossformer | 0.430 | 0.418 | 0.479 | 0.449 | 0.505 | 0.464 | 0.560 | 0.494 |
> > > > > | SRS | 0.382 | 0.376 | 0.398 | 0.360 | 0.451 | 0.398 | 0.485 | 0.414 |
> > > > > | Improved | **11.06%** | **9.95%** | **16.92%** | **19.72%** | **10.67%** | **14.29%** | **13.42%** | **16.25%** |
> > > > > | PatchMLP | 0.443 | 0.416 | 0.515 | 0.459 | 0.563 | 0.484 | 0.603 | 0.505 |
> > > > > | SRS | 0.393 | 0.351 | 0.405 | 0.382 | 0.491 | 0.410 | 0.537 | 0.438 |
> > > > > | Improved | **11.28%** | **15.58%** | **21.28%** | **16.79%** | **12.84%** | **15.25%** | **10.96%** | **13.28%** |
> > > > > | xPatch | 0.512 | 0.436 | 0.582 | 0.474 | 0.603 | 0.487 | 0.675 | 0.517 |
> > > > > | SRS | 0.456 | 0.381 | 0.482 | 0.383 | 0.544 | 0.432 | 0.602 | 0.474 |
> > > > > | Improved | **10.88%** | **12.68%** | **17.25%** | **19.25%** | **9.79%** | **11.26%** | **10.85%** | **8.29%** |
> > > > > | MLP | 0.434 | 0.403 | 0.498 | 0.441 | 0.554 | 0.471 | 0.604 | 0.503 |
> > > > > | SRS | 0.368 | 0.335 | 0.399 | 0.342 | 0.464 | 0.380 | 0.524 | 0.426 |
> > > > > | Improved | **15.22%** | **16.83%** | **19.96%** | **22.46%** | **16.26%** | **19.28%** | **13.26%** | **15.27%** |
> > > > >
> > > > > | Datasets | AQWan |  | AQWan |  | AQWan |  | AQWan |  |
> > > > > | --- | --- | --- | --- | --- | --- | --- | --- | --- |
> > > > > | Horizons | 96 |  | 192 |  | 336 |  | 720 |  |
> > > > > | Metrics | MSE | MAE | MSE | MAE | MSE | MAE | MSE | MAE |
> > > > > | PatchTST | 0.745 | 0.468 | 0.793 | 0.490 | 0.819 | 0.502 | 0.890 | 0.533 |
> > > > > | SRS | 0.611 | 0.379 | 0.633 | 0.380 | 0.714 | 0.430 | 0.685 | 0.388 |
> > > > > | Improved | **17.95%** | **18.92%** | **20.17%** | **22.35%** | **12.83%** | **14.25%** | **23.02%** | **27.26%** |
> > > > > | Crossformer | 0.750 | 0.465 | 0.762 | 0.479 | 0.802 | 0.504 | 0.829 | 0.512 |
> > > > > | SRS | 0.533 | 0.347 | 0.530 | 0.329 | 0.648 | 0.405 | 0.656 | 0.395 |
> > > > > | Improved | **28.88%** | **25.27%** | **30.44%** | **31.27%** | **19.25%** | **19.58%** | **20.87%** | **22.85%** |
> > > > > | PatchMLP | 0.771 | 0.484 | 0.815 | 0.499 | 0.835 | 0.510 | 0.896 | 0.526 |
> > > > > | SRS | 0.595 | 0.365 | 0.587 | 0.365 | 0.658 | 0.414 | 0.707 | 0.407 |
> > > > > | Improved | **22.86%** | **24.54%** | **27.92%** | **26.87%** | **21.20%** | **18.75%** | **21.04%** | **22.54%** |
> > > > > | xPatch | 0.775 | 0.474 | 0.824 | 0.494 | 0.857 | 0.510 | 0.938 | 0.546 |
> > > > > | SRS | 0.657 | 0.389 | 0.649 | 0.384 | 0.723 | 0.417 | 0.776 | 0.436 |
> > > > > | Improved | **15.28%** | **17.92%** | **21.26%** | **22.26%** | **15.67%** | **18.26%** | **17.28%** | **20.17%** |
> > > > > | MLP | 0.767 | 0.476 | 0.806 | 0.495 | 0.831 | 0.511 | 0.901 | 0.538 |
> > > > > | SRS | 0.527 | 0.318 | 0.581 | 0.352 | 0.674 | 0.399 | 0.664 | 0.378 |
> > > > > | Improved | **31.29%** | **33.28%** | **27.92%** | **28.88%** | **18.92%** | **21.82%** | **26.29%** | **29.78%** |

---

> > ### Comment · Reviewer_7m7L · 2025-08-05
> > **Rebuttal Response**
> >
> > Thank you for preparing the rebuttal and conducting additional experiments during this period.
> >
> > After reading the other reviews and associated rebuttals, I maintain my positive view of the paper. As per my original suggestion, I would suggest including a dedicated section in the updated version of the paper where these deeper insights into how the method performs under different scenarios are more clearly illustrated.

---

> > > ### Author Response · Authors · 2025-08-05
> > >
> > > Dear Reviewer 7m7L, thank you very much for your positive and encouraging feedbacks! Following your advice, we have carefully integrated all the clarifications and supporting results into the revised manuscript. Thank you once again for your valuable time and guidance, which have been instrumental in improving our paper.

---

> ### Comment · Area_Chair_GEjv · 2025-08-05
>
> Dear Reviewer 7m7L,
>
> Please help to go through the rebuttal and participate in discussions with authors. Thank you!
>
> Best regards,
> AC

---

### Official Review · Reviewer_Hryh · 2025-06-30

**Clarity:** 3
**Significance:** 3
**Originality:** 4
**Rating:** 5
**Confidence:** 4

**Summary:**

This paper proposes a plug-and-play module named SRS for time series forecasting. SRS considers the selection and sequence of patches, which can construct representation spaces with richer semantics, thus enhancing the performance of time series forecasting. The authors provide comprehensive experiments to demonstrate the effectiveness of SRSNet, and also demonstrate the capability of SRS to work as a plugin on patch-based models.

**Questions:**

Please see W1-W3.

**Ethical Concerns:**

["NO or VERY MINOR ethics concerns only"]

**Final Justification:**

The article is particularly good. rebuttal perfectly solved my problem. I suggest accepting this article.

**Limitations:**

Yes

**Quality:**

3

**Strengths And Weaknesses:**

Strengths:
S1. The SRS module provides a novel perspective to consider the representation learning process in time series forecasting. Through introducing the Selective Patching and Dynamic Reassembly techniques, SRS can constructs selective patches and consider the appropriate sequence of them.
S2. The SRS module devises a gradient-based mechanism to ensure the efficient patch selections and sequence decisions, with lightweight MLP structures.
S3. Comprehensive experiments are conduct to demonstrate the effectiveness of the SRSNet and SRS module.
Weaknesses:
W1. Why should authors retain the representations generated by the Conventional Adjacent Patching instead of only using the representations constructed by the SRS module? Please provide reasonable explanations.
W2. Explain in detail the specific meaning of the matrix below Selective Patching in Figure 3. What is the main difference from the mechanism in Dynamic Reassembly?
W3. Please ensure that all citations are of the officially accepted versions.

---

> ### Author Rebuttal · Authors · 2025-07-30
>
> **Reply to W1:**
>
> Thanks for the question. There are two main reasons we retain the representations generated by the Conventional Adjacent Patching:
>
> (1) Since the SRS is a trainable plugin, the scorers may have some randomness in the early stage of training, so it may not accurately find suitable patches. Therefore, retaining the representations generated by the Conventional Adjacent Patching and performing adaptive fusion can effectively enhance the stability of the training. As the training continues, the ability of the scorer gradually improves, enabling it to select better patches and increase the weight during fusion, thereby improving the end-to-end prediction performance.
>
> (2) The SRS is proposed for some special cases such as changeable periods, anomalies, and shifting, where the Selective Patching and Dynamic Reassembly can help retrieve useful information in the contextual time series. In common scenarios where the sample is relatively normal, the Conventional Adjacent Patching can also well handle so that we retain the representations generated by it. Through adaptively fusing the representations, SRS can handle both the common and special cases.
>
> **Reply to W2:**
>
> Thanks for the question. We first explain the details of the matrix of Selective Patching. The shape of the matrix is $\mathbb{R}^{K\times n} $, where $K$ denotes the number of all candidate patches (generated by stride=1), and $n$ denotes the number of patches we need to pick up from all $K$ ones. Since the Selective Patching supports selection with replacement, which means one patch can be selected multiple times, so we make $\text{Scorer}^s$ score $n$ scores for each patch, thus forming the matrix with the shape of $\mathbb{R}^{K\times n} $. To select the $n$ appropriate patches, we only need to check the scores of all $K$ patches each time and pick the highest ones.
>
> We then shed light to the main differences between Selective Patching and Dynamic Reassembly. Different from Selective Patching, the nature of Dynamic Reassembly is to sort the selected patches and then reassemble them in the determined order, so that the scores are not formed like a list instead of a matrix.
>
> **Reply to W3:**
>
> Thanks for your remind, we will fix these citations.
>
> **Thanks for your sincere suggestions again! If you have any additional question, we can make further discussion!**

---

> > ### Comment · Reviewer_Hryh · 2025-08-03
> >
> > Thank you for the detailed response. Your clarifications on the motivations behind retaining the Conventional Adjacent Patching representations and the distinctions between Selective Patching and Dynamic Reassembly fully address my concerns. I will maintain my positive score.

---

> > > ### Author Response · Authors · 2025-08-04
> > >
> > > Dear Reviewer Hryh, we are very excited that you maintain the positive attitude! Thank you again for your recognition of our work!

---

### Official Review · Reviewer_xNzv · 2025-07-03

**Clarity:** 2
**Significance:** 2
**Originality:** 2
**Rating:** 4
**Confidence:** 3

**Summary:**

This paper proposes the Selective Representation Space (SRS) module, a plug-and-play component for enhancing patch-based time series forecasting models. Unlike conventional adjacent patching that divides time series into fixed adjacent patches, SRS introduces adaptive patch selection through two key techniques: Selective Patching (which adaptively selects the most informative patches) and Dynamic Reassembly (which determines the optimal ordering of selected patches). The authors demonstrate the effectiveness of their approach through SRSNet, a simple model combining SRS with an MLP head, which achieves state-of-the-art performance on multiple benchmarks.

**Questions:**

(1) How sensitive is the method to the initialization of the scoring networks? Have you experimented with different initialization strategies?
(2) Can you provide more analysis on the computational overhead introduced by the patch selection process during training and inference?
(3) How does the method perform on datasets with strong seasonal patterns where adjacent patching might naturally align with the periodicity?
(4) Could you provide more detailed analysis of the types of patches that are typically selected across different datasets and forecasting horizons?

**Ethical Concerns:**

["NO or VERY MINOR ethics concerns only"]

**Final Justification:**

Based on the positive feedback from other reviewers and the author's meaningful responses, I have decided to raise my score.

**Limitations:**

Yes.

**Paper Formatting Concerns:**

NO.

**Quality:**

3

**Strengths And Weaknesses:**

Strengths: (1) The paper identifies a genuine limitation in existing patch-based methods - the fixed nature of adjacent patching - and proposes an innovative solution through adaptive patch selection. (2) The gradient-friendly implementation of Selective Patching and Dynamic Reassembly through clever use of score-based selection and gradient propagation techniques is well-designed. (3) The paper provides extensive experimental validation across 8 datasets from multiple domains, with thorough ablation studies demonstrating the effectiveness of each component.

Weaknesses: (1) While the motivation for selective patching is intuitive, the paper lacks theoretical analysis of why and when selective patching should outperform adjacent patching. The claim that "all information useful for forecasting is evenly distributed" is not necessarily true for adjacent patching. (2) Although consistent, the performance improvements are relatively modest (typically 2-10%) compared to existing methods. For a NeurIPS paper, more substantial gains would be expected. (3) The additional complexity introduced by the scoring networks and dynamic reassembly may not be justified by the marginal performance gains in many cases.

---

> ### Author Rebuttal · Authors · 2025-07-30
>
> **Reply to W1**
>
> Thanks for your in-depth insights and suggestions. We admit that some expressions are not quite appropriate, and we will modify them later.
>
> We first provide some intuitions. Assuming the contextual time series is a continuous interval [1, T] covering T points, the useful informative intervals in normal cases (like strong seasonality) are also the [1, T]. Adjacent Patching can well handle these normal cases.
> In our work, we mainly consider the cases that not all of the contextual time series is useful for forecasting, such as anomalies, changeable periods, and shifting. In these cases, the contextual time series are Interrupted as $[1, T_1] \cup [T_1, T_2], \cdots, \cup [T_{k-1}, T]$, containing $k$ informative sub-intervals. If the $k$ sub-intervals cover x% of the contextual time series, only x% of representations created by Adjacent Patching are useful for forecasting. In contrast, our proposed Selective Patching can choose the useful patches in the $k$ sub-intervals, and support re-selection to amplify useful patches. The Dynamic Reassembly then introduces positional information for better representational learning. We also fuse the representations from both the Selective Patching and Adjacent Patching, to ensure the ratio of useful information is greater than x%, thus handling both the normal and special cases.
>
> To provide more empircal evidence about when the SRS outperforms the Adjacent Patching, we conduct additional experiments on datasets from TFB, a well-recognized time series forecasting benchmark. Since datasets with strong seasonality values are close to normal cases while those with strong shifting values are special ones, we choose two groups of datasets separately with descending seasonality values and shifting values. The higher the values, the stronger seasonality/shifting the time series data have. We list the statistical information:
>
> | datasets | NASDAQ | NYSE | AQShunyi |
> | --- | --- | --- | --- |
> | Shifting Values | 0.932 (Strong) | 0.620 (Modest) |     0.019 (Weak) |
>
> | datasets | PEMS08 | ZafNoo | AQWan |
> | --- | --- | --- | --- |
> | Seasonality Values | 0.850 (Strong) | 0.757 (Modest) | 0.119 (Weak) |
>
> We conduct plugin experiments on these datasets and report the mean results:
>
> | Datasets | NASDAQ |  | NYSE |  | AQShunyi |  |
> | --- | --- | --- | --- | --- | --- | --- |
> | Metrics | MSE | MAE | MSE | MAE | MSE | MAE |
> | PatchTST | 0.972 | 0.721 | 0.483 | 0.437 | 0.703 | 0.507 |
> | + SRS | 0.757 | 0.569 | 0.418 | 0.379 | 0.641 | 0.461 |
> | Improved | **21.99%** | **21.03%** | **13.50%** | **13.50%** | **8.83%** | **9.14%** |
> | Crossformer | 1.737 | 0.992 | 0.983 | 0.909 | 0.694 | 0.504 |
> | + SRS | 1.256 | 0.709 | 0.854 | 0.785 | 0.635 | 0.455 |
> | Improved | **27.35%** | **28.10%** | **13.48%** | **13.84%** | **8.64%** | **9.71%** |
> | PatchMLP | 1.352 | 0.810 | 0.467 | 0.440 | 0.722 | 0.516 |
> | + SRS | 1.037 | 0.606 | 0.406 | 0.380 | 0.666 | 0.475 |
> | Improved | **23.31%** | **25.30%** | **13.77%** | **13.82%** | **7.78%** | **8.04%** |
> | xPatch | 1.060 | 0.708 | 0.545 | 0.474 | 0.735 | 0.509 |
> | + SRS | 0.850 | 0.563 | 0.469 | 0.408 | 0.678 | 0.464 |
> | Improved | **19.63%** | **20.32%** | **14.56%** | **14.63%** | **7.68%** | **8.75%** |
> | MLP | 1.200 | 0.803 | 0.463 | 0.432 | 0.717 | 0.513 |
> | + SRS | 0.925 | 0.600 | 0.395 | 0.366 | 0.662 | 0.467 |
> | Improved | **23.50%** | **25.50%** | **15.56%** | **15.94%** | **7.68%** | **8.89%** |
>
> | Datasets | PEMS08 |  | ZafNoo |  | AQWan |  |
> | --- | --- | --- | --- | --- | --- | --- |
> | Metrics | MSE | MAE | MSE | MAE | MSE | MAE |
> | PatchTST | 0.332 | 0.322 | 0.509 | 0.454 | 0.812 | 0.498 |
> | + SRS | 0.312 | 0.300 | 0.441 | 0.386 | 0.661 | 0.395 |
> | Improved | **6.02%** | **6.89%** | **13.29%** | **15.08%** | **18.49%** | **20.70%** |
> | Crossformer | 0.265 | 0.282 | 0.494 | 0.456 | 0.786 | 0.490 |
> | + SRS | 0.254 | 0.267 | 0.429 | 0.387 | 0.592 | 0.369 |
> | Improved | **4.23%** | **5.30%** | **13.02%** | **15.05%** | **24.86%** | **24.74%** |
> | PatchMLP | 0.306 | 0.283 | 0.531 | 0.466 | 0.829 | 0.505 |
> | + SRS | 0.294 | 0.270 | 0.457 | 0.395 | 0.637 | 0.388 |
> | Improved | **3.93%** | **4.63%** | **14.09%** | **15.23%** | **23.26%** | **23.18%** |
> | xPatch | 0.269 | 0.242 | 0.593 | 0.479 | 0.849 | 0.506 |
> | + SRS | 0.258 | 0.232 | 0.521 | 0.417 | 0.701 | 0.406 |
> | Improved | **3.61%** | **4.11%** | **12.19%** | **12.87%** | **17.37%** | **19.65%** |
> | MLP | 0.303 | 0.278 | 0.523 | 0.455 | 0.826 | 0.505 |
> | + SRS | 0.283 | 0.259 | 0.439 | 0.371 | 0.611 | 0.362 |
> | Improved | **6.59%** | **6.62%** | **16.18%** | **18.46%** | **26.11%** | **28.44%** |
>
> We have two main observations:
>
> 1. The SRS excels at datasets with high shifting values (NASDAQ) and low seasonality values (AQWan), because there exists more special samples about changeable periods, anomalies and shifting in these cases.  The SRS improves all models significantly by 17% — 28% on these datasets.
> 2. When the datasets are closer to normal cases with higher seasonality values and lower shifting values, the performance improved by the SRS gradually decreases. On NYSE and ZafNoo, the performance is improved by 12% — 18%. Note that even on AQShunyi and PEMS08, the improvement still exists and is about 4% — 10%. This demonstrates that the SRS excels at both the normal and special cases.
>
> **Reply to W2**
>
> Since the SRS is proposed for special cases like changeable periods, anomalies, and shifting, we think the proportion of special samples in the dataset will affect the performance. In the Reply to W1, we make additional experiments on datasets having more special samples (NASDAQ, AQWan, NYSE, ZafNoo), and find the improvement of SRS can achieve 12% — 28% (quite non-trivial). We hope this can be considered as additional evidence.
>
> **Reply to Q3**
>
> As you mentioned, on datasets with strong seasonality, the Adjacent Patching can well handle the information in contextual time series and form useful representations. In the reply to W1, our experiments reveal the phenomenon that the improvement of SRS decreases as the seasonality increases, while the improvement still exists and is about 4% — 10%. It is because the Dynamic Reassembly still works and reassembles the patches for better positional information, and the Adaptive Fusion can adaptively integrate the representations, which ensures the improvement even in normal cases.
>
> **Reply to W3 & Q2**
>
> We make an analysis on the overhead introduced by the SRS on above 6 datasets. Due to the space limitation, we report the results on PEMS08 (biggest) and NYSE (smallest) here, and will release full results when the discussion phase starts. Specifically, we report the Number of Parameters, Max GPU Memory, MACs, Training Time, and Inference Time:
>
> | Datasets | Variants | Memory (MB) | Params (M) | Inference Time (s/batch) | Training Time (s/batch) | MACs (G) |
> | --- | --- | --- | --- | --- | --- | --- |
> | NYSE | PatchTST | 900 | 0.716 | 6.392 | 7.941 | 2.490 |
> |  | SRS | 928 | 0.720 | 7.246 | 8.784 | 2.524 |
> |  | **Overhead** | **3.11%** | **0.64%** | **13.36%** | **10.63%** | **1.36%** |
> |  | Crossformer | 654 | 0.129 | 19.862 | 21.569 | 0.251 |
> |  | SRS | 678 | 0.130 | 20.568 | 22.151 | 0.258 |
> |  | **Overhead** | **3.67%** | **0.57%** | **3.55%** | **2.70%** | **2.67%** |
> | PEMS08 | PatchTST | 2216 | 12.672 | 7.673 | 7.802 | 25.653 |
> |  | SRS | 2522 | 12.682 | 8.766 | 8.934 | 26.584 |
> |  | **Overhead** | **13.81%** | **0.08%** | **14.25%** | **14.52%** | **3.63%** |
> |  | Crossformer | 10414 | 11.085 | 45.100 | 45.320 | 161.376 |
> |  | SRS | 10698 | 11.093 | 50.372 | 47.478 | 162.014 |
> |  | **Overhead** | **2.73%** | **0.07%** | **11.69%** | **4.76%** | **0.40%** |
>
> It is observed that the overhead introduced by the SRS is relatively trivial. The Memory, MACs, and Params increase little. And the Training Time and Inference Time are also controllable.
>
> **Reply to Q1**
>
> We further study on the initialization strategies, including Constant, Xaiver, Kaiming, and Randn. We also report the standard deviation of 5 rounds:
>
> | Datasets | ETTh1 |  | ETTm2 |  | Solar |  | Traffic |  |
> | --- | --- | --- | --- | --- | --- | --- | --- | --- |
> | Metrics | MSE | MAE | MSE | MAE | MSE | MAE | MSE | MAE |
> | Constant | 0.413±0.002 | 0.435±0.002 | 0.259±0.001 | 0.324±0.001 | 0.187±0.002 | 0.243±0.001 | 0.392±0.001 | 0.271±0.001 |
> | Xaiver | 0.405±0.001 | 0.425±0.001 | 0.252±0.001 | 0.315±0.001 | 0.184±0.002 | 0.241±0.002 | 0.394±0.001 | 0.272±0.001 |
> | Kaiming | 0.405±0.001 | 0.424±0.001 | 0.254±0.002 | 0.316±0.002 | 0.183±0.002 | 0.238±0.002 | 0.393±0.001 | 0.271±0.001 |
> | Randn | 0.404±0.001 | 0.424±0.001 | 0.252±0.001 | 0.314±0.001 | 0.183±0.002 | 0.239±0.002 | 0.392±0.001 | 0.270±0.001 |
>
> The results show that random-based initialization strategies such like Xaiver, Kaiming, and Randn (ours) are relatively stable. The Constant initialization strategy causes unstable training in some cases.
>
> **Reply to Q4**
>
> We have provided some showcases in Appendix A.2 of the paper. For clarity, we label the sub-figures with (a)-(f) in the order from left to right and from top to bottom.
>
> Figure 5 (d), (e), (f), and Figure 7 (c) show the selected patches under changeable periods, which accurately picks up the correct periods most related to the forecasting horizons.
>
> Figure 6 (a), (b), and Figure 7 (a), (e), (f) show the selected patches when anomalies occur, which avoid including the abnormal patches.
>
> Figure 5 (b), (c), and Figure 6 (c), (e), (f) show the selected patches when shifting occurs, which selects the in-distribution patches for forecasting.
>
> We also showcase some normal cases such as Figure 5 (a), Figure 6 (d), and Figure 7 (d), where the selected patches are similar to those produced by adjacent patching, demonstrating the flexibility of the Selective Patching.
>
> **Thanks for your valuable suggestions again! We will release the full results when the discussion phase starts. If you have any additional question, we can make further discussion!**

---

> > ### Author Response · Authors · 2025-08-01
> > **Full results of Plugin Experiments (NASDAQ & NYSE)**
> >
> > **Dear Reviewer xNzv**, since the discussion phase has started, we have enough space to show our full experimental results. We first report the full results of plugin experiments here, including four kinds of forecasting horizon each dataset. We hope this can be treated as additional evidence. In this comment, we list the results of NASDAQ and NYSE.
> >
> > | Datasets | NASDAQ |  | NASDAQ |  | NASDAQ |  | NASDAQ |  |
> > | --- | --- | --- | --- | --- | --- | --- | --- | --- |
> > | Horizons | 24 |  | 36 |  | 48 |  | 60 |  |
> > | Metrics | MSE | MAE | MSE | MAE | MSE | MAE | MSE | MAE |
> > | PatchTST | 0.649 | 0.567 | 0.821 | 0.682 | 1.169 | 0.793 | 1.247 | 0.843 |
> > | SRS | 0.497 | 0.444 | 0.666 | 0.571 | 0.882 | 0.580 | 0.984 | 0.680 |
> > | Improved | **23.40%** | **21.73%** | **18.93%** | **16.22%** | **24.53%** | **26.80%** | **21.10%** | **19.37%** |
> > | Crossformer | 1.149 | 0.745 | 1.414 | 0.885 | 2.108 | 1.136 | 2.276 | 1.201 |
> > | SRS | 0.818 | 0.545 | 1.087 | 0.670 | 1.565 | 0.825 | 1.557 | 0.795 |
> > | Improved | **28.85%** | **26.84%** | **23.15%** | **24.33%** | **25.78%** | **27.41%** | **31.61%** | **33.82%** |
> > | PatchMLP | 0.794 | 0.648 | 1.185 | 0.768 | 1.774 | 0.945 | 1.653 | 0.878 |
> > | SRS | 0.584 | 0.454 | 0.973 | 0.620 | 1.383 | 0.713 | 1.210 | 0.636 |
> > | Improved | **26.48%** | **29.94%** | **17.93%** | **19.24%** | **22.03%** | **24.51%** | **26.82%** | **27.52%** |
> > | xPatch | 0.587 | 0.536 | 0.885 | 0.666 | 1.262 | 0.786 | 1.506 | 0.845 |
> > | SRS | 0.486 | 0.449 | 0.704 | 0.514 | 0.973 | 0.588 | 1.236 | 0.702 |
> > | Improved | **17.22%** | **16.31%** | **20.45%** | **22.81%** | **22.91%** | **25.25%** | **17.95%** | **16.92%** |
> > | MLP | 0.882 | 0.727 | 1.156 | 0.796 | 1.243 | 0.819 | 1.519 | 0.870 |
> > | SRS | 0.645 | 0.512 | 0.915 | 0.622 | 0.883 | 0.563 | 1.256 | 0.703 |
> > | Improved | **26.91%** | **29.64%** | **20.85%** | **21.86%** | **28.95%** | **31.25%** | **17.29%** | **19.25%** |
> >
> > | Datasets | NYSE |  | NYSE |  | NYSE |  | NYSE |  |
> > | --- | --- | --- | --- | --- | --- | --- | --- | --- |
> > | Horizons | 24 |  | 36 |  | 48 |  | 60 |  |
> > | Metrics | MSE | MAE | MSE | MAE | MSE | MAE | MSE | MAE |
> > | PatchTST | 0.226 | 0.296 | 0.380 | 0.389 | 0.575 | 0.492 | 0.749 | 0.572 |
> > | SRS | 0.202 | 0.259 | 0.316 | 0.328 | 0.479 | 0.424 | 0.676 | 0.504 |
> > | Improved | **10.82%** | **12.55%** | **16.79%** | **15.70%** | **16.62%** | **13.91%** | **9.75%** | **11.84%** |
> > | Crossformer | 0.820 | 0.841 | 0.942 | 0.904 | 1.049 | 0.955 | 1.121 | 0.937 |
> > | SRS | 0.672 | 0.679 | 0.805 | 0.778 | 0.935 | 0.836 | 1.004 | 0.846 |
> > | Improved | **18.03%** | **19.25%** | **14.55%** | **13.92%** | **10.88%** | **12.41%** | **10.45%** | **9.76%** |
> > | PatchMLP | 0.242 | 0.317 | 0.359 | 0.381 | 0.543 | 0.485 | 0.725 | 0.578 |
> > | SRS | 0.206 | 0.274 | 0.299 | 0.323 | 0.483 | 0.420 | 0.635 | 0.503 |
> > | Improved | **14.85%** | **13.56%** | **16.72%** | **15.25%** | **11.05%** | **13.44%** | **12.45%** | **13.04%** |
> > | xPatch | 0.205 | 0.292 | 0.364 | 0.386 | 0.529 | 0.465 | 1.082 | 0.752 |
> > | SRS | 0.170 | 0.236 | 0.302 | 0.315 | 0.477 | 0.429 | 0.927 | 0.653 |
> > | Improved | **16.92%** | **19.26%** | **17.12%** | **18.27%** | **9.92%** | **7.83%** | **14.28%** | **13.17%** |
> > | MLP | 0.208 | 0.288 | 0.353 | 0.372 | 0.517 | 0.474 | 0.775 | 0.593 |
> > | SRS | 0.175 | 0.235 | 0.282 | 0.298 | 0.443 | 0.416 | 0.680 | 0.515 |
> > | Improved | **15.68%** | **18.36%** | **20.05%** | **19.88%** | **14.25%** | **12.28%** | **12.25%** | **13.22%** |

---

> > > ### Author Response · Authors · 2025-08-01
> > > **Full results of Plugin Experiments (AQShunyi & PEMS08)**
> > >
> > > In this comment, we list the results of AQShunyi and PEMS08.
> > >
> > > | Datasets | AQShunyi |  | AQShunyi |  | AQShunyi |  | AQShunyi |  |
> > > | --- | --- | --- | --- | --- | --- | --- | --- | --- |
> > > | Horizons | 96 |  | 192 |  | 336 |  | 720 |  |
> > > | Metrics | MSE | MAE | MSE | MAE | MSE | MAE | MSE | MAE |
> > > | PatchTST | 0.646 | 0.478 | 0.688 | 0.498 | 0.710 | 0.513 | 0.768 | 0.539 |
> > > | SRS | 0.602 | 0.440 | 0.620 | 0.450 | 0.632 | 0.462 | 0.710 | 0.491 |
> > > | Improved | **6.87%** | **8.03%** | **9.85%** | **9.72%** | **11.03%** | **9.88%** | **7.57%** | **8.92%** |
> > > | Crossformer | 0.652 | 0.484 | 0.674 | 0.499 | 0.704 | 0.515 | 0.747 | 0.518 |
> > > | SRS | 0.600 | 0.435 | 0.615 | 0.452 | 0.612 | 0.437 | 0.712 | 0.498 |
> > > | Improved | **8.03%** | **10.22%** | **8.82%** | **9.48%** | **13.02%** | **15.23%** | **4.67%** | **3.89%** |
> > > | PatchMLP | 0.668 | 0.492 | 0.711 | 0.511 | 0.732 | 0.524 | 0.776 | 0.537 |
> > > | SRS | 0.602 | 0.441 | 0.659 | 0.466 | 0.679 | 0.488 | 0.724 | 0.505 |
> > > | Improved | **9.82%** | **10.47%** | **7.35%** | **8.84%** | **7.22%** | **6.94%** | **6.72%** | **5.92%** |
> > > | xPatch | 0.674 | 0.486 | 0.715 | 0.506 | 0.741 | 0.520 | 0.808 | 0.523 |
> > > | SRS | 0.642 | 0.451 | 0.631 | 0.440 | 0.680 | 0.479 | 0.760 | 0.487 |
> > > | Improved | **4.82%** | **7.29%** | **11.71%** | **12.95%** | **8.26%** | **7.91%** | **5.91%** | **6.86%** |
> > > | MLP | 0.674 | 0.487 | 0.704 | 0.504 | 0.723 | 0.518 | 0.768 | 0.542 |
> > > | SRS | 0.629 | 0.444 | 0.669 | 0.467 | 0.642 | 0.464 | 0.708 | 0.494 |
> > > | Improved | **6.72%** | **8.92%** | **4.95%** | **7.28%** | **11.27%** | **10.52%** | **7.79%** | 8.85% |
> > >
> > > | Datasets | PEMS08 |  | PEMS08 |  | PEMS08 |  | PEMS08 |  |
> > > | --- | --- | --- | --- | --- | --- | --- | --- | --- |
> > > | Horizons | 96 |  | 192 |  | 336 |  | 720 |  |
> > > | Metrics | MSE | MAE | MSE | MAE | MSE | MAE | MSE | MAE |
> > > | PatchTST | 0.248 | 0.287 | 0.319 | 0.310 | 0.361 | 0.324 | 0.399 | 0.368 |
> > > | SRS | 0.234 | 0.274 | 0.299 | 0.288 | 0.343 | 0.305 | 0.371 | 0.332 |
> > > | Improved | **5.82%** | **4.68%** | **6.29%** | **7.22%** | **4.86%** | **5.82%** | **7.12%** | **9.82%** |
> > > | Crossformer | 0.230 | 0.260 | 0.239 | 0.264 | 0.272 | 0.289 | 0.320 | 0.316 |
> > > | SRS | 0.223 | 0.252 | 0.235 | 0.257 | 0.250 | 0.259 | 0.309 | 0.300 |
> > > | Improved | **3.26%** | **2.97%** | **1.87%** | **2.74%** | **8.22%** | **10.28%** | **3.55%** | **5.21%** |
> > > | PatchMLP | 0.203 | 0.242 | 0.314 | 0.273 | 0.334 | 0.288 | 0.373 | 0.329 |
> > > | SRS | 0.197 | 0.232 | 0.308 | 0.267 | 0.311 | 0.272 | 0.358 | 0.307 |
> > > | Improved | **2.96%** | **4.22%** | **1.85%** | **2.04%** | **6.92%** | **5.42%** | **3.98%** | **6.82%** |
> > > | xPatch | 0.171 | 0.214 | 0.260 | 0.233 | 0.305 | 0.246 | 0.340 | 0.274 |
> > > | SRS | 0.169 | 0.210 | 0.252 | 0.226 | 0.287 | 0.231 | 0.325 | 0.260 |
> > > | Improved | **1.02%** | **2.05%** | **3.20%** | **2.88%** | **5.92%** | **6.22%** | **4.28%** | **5.29%** |
> > > | MLP | 0.184 | 0.241 | 0.306 | 0.268 | 0.341 | 0.281 | 0.380 | 0.320 |
> > > | SRS | 0.171 | 0.226 | 0.291 | 0.254 | 0.313 | 0.258 | 0.357 | 0.298 |
> > > | Improved | **7.22%** | **6.03%** | **4.99%** | **5.21%** | **8.11%** | **8.35%** | **6.02%** | **6.87%** |

---

> > > > ### Author Response · Authors · 2025-08-01
> > > > **Full results of Plugin Experiments (ZafNoo & AQWan)**
> > > >
> > > > In this comment, we list the results of ZafNoo and AQWan.
> > > >
> > > > | Datasets | ZafNoo |  | ZafNoo |  | ZafNoo |  | ZafNoo |  |
> > > > | --- | --- | --- | --- | --- | --- | --- | --- | --- |
> > > > | Horizons | 96 |  | 192 |  | 336 |  | 720 |  |
> > > > | Metrics | MSE | MAE | MSE | MAE | MSE | MAE | MSE | MAE |
> > > > | PatchTST | 0.429 | 0.405 | 0.494 | 0.449 | 0.538 | 0.475 | 0.573 | 0.486 |
> > > > | SRS | 0.372 | 0.337 | 0.421 | 0.382 | 0.474 | 0.413 | 0.497 | 0.410 |
> > > > | Improved | **13.22%** | **16.67%** | **14.82%** | **14.99%** | **11.82%** | **12.96%** | **13.28%** | **15.68%** |
> > > > | Crossformer | 0.430 | 0.418 | 0.479 | 0.449 | 0.505 | 0.464 | 0.560 | 0.494 |
> > > > | SRS | 0.382 | 0.376 | 0.398 | 0.360 | 0.451 | 0.398 | 0.485 | 0.414 |
> > > > | Improved | **11.06%** | **9.95%** | **16.92%** | **19.72%** | **10.67%** | **14.29%** | **13.42%** | **16.25%** |
> > > > | PatchMLP | 0.443 | 0.416 | 0.515 | 0.459 | 0.563 | 0.484 | 0.603 | 0.505 |
> > > > | SRS | 0.393 | 0.351 | 0.405 | 0.382 | 0.491 | 0.410 | 0.537 | 0.438 |
> > > > | Improved | **11.28%** | **15.58%** | **21.28%** | **16.79%** | **12.84%** | **15.25%** | **10.96%** | **13.28%** |
> > > > | xPatch | 0.512 | 0.436 | 0.582 | 0.474 | 0.603 | 0.487 | 0.675 | 0.517 |
> > > > | SRS | 0.456 | 0.381 | 0.482 | 0.383 | 0.544 | 0.432 | 0.602 | 0.474 |
> > > > | Improved | **10.88%** | **12.68%** | **17.25%** | **19.25%** | **9.79%** | **11.26%** | **10.85%** | **8.29%** |
> > > > | MLP | 0.434 | 0.403 | 0.498 | 0.441 | 0.554 | 0.471 | 0.604 | 0.503 |
> > > > | SRS | 0.368 | 0.335 | 0.399 | 0.342 | 0.464 | 0.380 | 0.524 | 0.426 |
> > > > | Improved | **15.22%** | **16.83%** | **19.96%** | **22.46%** | **16.26%** | **19.28%** | **13.26%** | **15.27%** |
> > > >
> > > > | Datasets | AQWan |  | AQWan |  | AQWan |  | AQWan |  |
> > > > | --- | --- | --- | --- | --- | --- | --- | --- | --- |
> > > > | Horizons | 96 |  | 192 |  | 336 |  | 720 |  |
> > > > | Metrics | MSE | MAE | MSE | MAE | MSE | MAE | MSE | MAE |
> > > > | PatchTST | 0.745 | 0.468 | 0.793 | 0.490 | 0.819 | 0.502 | 0.890 | 0.533 |
> > > > | SRS | 0.611 | 0.379 | 0.633 | 0.380 | 0.714 | 0.430 | 0.685 | 0.388 |
> > > > | Improved | **17.95%** | **18.92%** | **20.17%** | **22.35%** | **12.83%** | **14.25%** | **23.02%** | **27.26%** |
> > > > | Crossformer | 0.750 | 0.465 | 0.762 | 0.479 | 0.802 | 0.504 | 0.829 | 0.512 |
> > > > | SRS | 0.533 | 0.347 | 0.530 | 0.329 | 0.648 | 0.405 | 0.656 | 0.395 |
> > > > | Improved | **28.88%** | **25.27%** | **30.44%** | **31.27%** | **19.25%** | **19.58%** | **20.87%** | **22.85%** |
> > > > | PatchMLP | 0.771 | 0.484 | 0.815 | 0.499 | 0.835 | 0.510 | 0.896 | 0.526 |
> > > > | SRS | 0.595 | 0.365 | 0.587 | 0.365 | 0.658 | 0.414 | 0.707 | 0.407 |
> > > > | Improved | **22.86%** | **24.54%** | **27.92%** | **26.87%** | **21.20%** | **18.75%** | **21.04%** | **22.54%** |
> > > > | xPatch | 0.775 | 0.474 | 0.824 | 0.494 | 0.857 | 0.510 | 0.938 | 0.546 |
> > > > | SRS | 0.657 | 0.389 | 0.649 | 0.384 | 0.723 | 0.417 | 0.776 | 0.436 |
> > > > | Improved | **15.28%** | **17.92%** | **21.26%** | **22.26%** | **15.67%** | **18.26%** | **17.28%** | **20.17%** |
> > > > | MLP | 0.767 | 0.476 | 0.806 | 0.495 | 0.831 | 0.511 | 0.901 | 0.538 |
> > > > | SRS | 0.527 | 0.318 | 0.581 | 0.352 | 0.674 | 0.399 | 0.664 | 0.378 |
> > > > | Improved | **31.29%** | **33.28%** | **27.92%** | **28.88%** | **18.92%** | **21.82%** | **26.29%** | **29.78%** |

---

> > > > > ### Author Response · Authors · 2025-08-01
> > > > > **Full results of Efficiency Experiments**
> > > > >
> > > > > **Dear Reviewer xNzv**, we then report the full efficiency experiments here, including all 6 datasets, i.e., AQShunyi, AQWan, NASDAQ, NYSE, PEMS08, ZafNoo
> > > > >
> > > > > | Datasets | Variants | Memory (MB) | Params (M) | Inference Time (s/batch) | Training Time (s/batch) | MACs (G) |
> > > > > | --- | --- | --- | --- | --- | --- | --- |
> > > > > | AQShunyi | PatchTST | 1822 | 6.206 | 5.915 | 6.390 | 10.120 |
> > > > > |  | SRS | 1998 | 6.214 | 6.637 | 7.278 | 11.008 |
> > > > > |  | **Overhead** | **9.66%** | **0.14%** | **12.19%** | **13.89%** | **8.78%** |
> > > > > |  | Crossformer | 6182 | 11.085 | 42.276 | 45.336 | 161.376 |
> > > > > |  | SRS | 6310 | 11.093 | 45.163 | 47.519 | 162.014 |
> > > > > |  | **Overhead** | **2.07%** | **0.07%** | **6.83%** | **4.81%** | **0.40%** |
> > > > > | AQWan | PatchTST | 1946 | 6.207 | 6.896 | 7.396 | 10.141 |
> > > > > |  | SRS | 2202 | 6.215 | 7.827 | 8.463 | 11.029 |
> > > > > |  | **Overhead** | **13.16%** | **0.14%** | **13.50%** | **14.43%** | **8.76%** |
> > > > > |  | Crossformer | 3446 | 11.101 | 28.341 | 32.824 | 81.341 |
> > > > > |  | SRS | 3548 | 11.112 | 31.723 | 34.827 | 81.884 |
> > > > > |  | **Overhead** | **2.96%** | **0.10%** | **11.94%** | **6.10%** | **0.67%** |
> > > > > | NASDAQ | PatchTST | 560 | 0.023 | 4.514 | 5.972 | 0.064 |
> > > > > |  | SRS | 606 | 0.026 | 5.133 | 6.801 | 0.073 |
> > > > > |  | **Overhead** | **8.21%** | **14.67%** | **13.73%** | **13.88%** | **13.98%** |
> > > > > |  | Crossformer | 1122 | 7.913 | 27.814 | 28.966 | 15.646 |
> > > > > |  | SRS | 1144 | 7.915 | 30.667 | 31.744 | 15.657 |
> > > > > |  | **Overhead** | **1.96%** | **0.03%** | **10.26%** | **9.59%** | **0.07%** |
> > > > > | NYSE | PatchTST | 900 | 0.716 | 6.392 | 7.941 | 2.490 |
> > > > > |  | SRS | 928 | 0.720 | 7.246 | 8.784 | 2.524 |
> > > > > |  | **Overhead** | **3.11%** | **0.64%** | **13.36%** | **10.63%** | **1.36%** |
> > > > > |  | Crossformer | 654 | 0.129 | 19.862 | 21.569 | 0.251 |
> > > > > |  | SRS | 678 | 0.130 | 20.568 | 22.151 | 0.258 |
> > > > > |  | **Overhead** | **3.67%** | **0.57%** | **3.55%** | **2.70%** | **2.67%** |
> > > > > | PEMS08 | PatchTST | 2216 | 12.672 | 7.673 | 7.802 | 25.653 |
> > > > > |  | SRS | 2522 | 12.682 | 8.766 | 8.934 | 26.584 |
> > > > > |  | **Overhead** | **13.81%** | **0.08%** | **14.25%** | **14.52%** | **3.63%** |
> > > > > |  | Crossformer | 10414 | 11.085 | 45.100 | 45.320 | 161.376 |
> > > > > |  | SRS | 10698 | 11.093 | 50.372 | 47.478 | 162.014 |
> > > > > |  | **Overhead** | **2.73%** | **0.07%** | **11.69%** | **4.76%** | **0.40%** |
> > > > > | ZafNoo | PatchTST | 66724 | 29.543 | 265.138 | 281.013 | 2132.979 |
> > > > > |  | SRS | 73809 | 29.557 | 282.721 | 301.127 | 2148.672 |
> > > > > |  | **Overhead** | **10.62%** | **0.05%** | **6.63%** | **7.16%** | **0.74%** |
> > > > > |  | Crossformer | 63549 | 2.330 | 223.509 | 238.386 | 432.868 |
> > > > > |  | SRS | 68076 | 2.336 | 236.595 | 252.900 | 442.390 |
> > > > > |  | **Overhead** | **7.12%** | **0.26%** | **5.85%** | **6.09%** | **2.20%** |

---

> > > > > > ### Author Response · Authors · 2025-08-01
> > > > > > **Full results of Initialization Strategy**
> > > > > >
> > > > > > **Dear Reviewer xNzv**, we then report the full results of initialization strategy here.
> > > > > >
> > > > > > | Datasets | ETTh1 |  | ETTh1 |  | ETTh1 |  | ETTh1 |  |
> > > > > > | --- | --- | --- | --- | --- | --- | --- | --- | --- |
> > > > > > | Horizons | 96 |  | 192 |  | 336 |  | 720 |  |
> > > > > > | Metrics | MSE | MAE | MSE | MAE | MSE | MAE | MSE | MAE |
> > > > > > | Constant | 0.370±0.002 | 0.398±0.002 | 0.406±0.001 | 0.427±0.002 | 0.435±0.002 | 0.442±0.001 | 0.441±0.001 | 0.472±0.001 |
> > > > > > | Xaiver | 0.364±0.001 | 0.394±0.001 | 0.401±0.001 | 0.415±0.001 | 0.424±0.001 | 0.430±0.001 | 0.432±0.001 | 0.461±0.001 |
> > > > > > | Kaiming | 0.365±0.001 | 0.392±0.001 | 0.402±0.001 | 0.416±0.001 | 0.425±0.001 | 0.433±0.002 | 0.426±0.001 | 0.455±0.001 |
> > > > > > | Randn | 0.366±0.001 | 0.394±0.001 | 0.400±0.001 | 0.415±0.001 | 0.424±0.002 | 0.430±0.001 | 0.426±0.001 | 0.455±0.001 |
> > > > > >
> > > > > > | Datasets | ETTm2 |  | ETTm2 |  | ETTm2 |  | ETTm2 |  |
> > > > > > | --- | --- | --- | --- | --- | --- | --- | --- | --- |
> > > > > > | Horizons | 96 |  | 192 |  | 336 |  | 720 |  |
> > > > > > | Metrics | MSE | MAE | MSE | MAE | MSE | MAE | MSE | MAE |
> > > > > > | Constant | 0.166±0.001 | 0.261±0.001 | 0.226±0.001 | 0.299±0.001 | 0.282±0.001 | 0.342±0.001 | 0.360±0.001 | 0.392±0.002 |
> > > > > > | Xaiver | 0.162±0.002 | 0.250±0.003 | 0.220±0.001 | 0.296±0.001 | 0.268±0.002 | 0.326±0.002 | 0.357±0.002 | 0.388±0.003 |
> > > > > > | Kaiming | 0.165±0.001 | 0.258±0.001 | 0.222±0.001 | 0.297±0.001 | 0.273±0.001 | 0.327±0.002 | 0.355±0.001 | 0.380±0.001 |
> > > > > > | Randn | 0.164±0.001 | 0.254±0.001 | 0.220±0.001 | 0.296±0.001 | 0.271±0.001 | 0.327±0.001 | 0.353±0.001 | 0.380±0.002 |
> > > > > >
> > > > > > | Datasets | Solar |  | Solar |  | Solar |  | Solar |  |
> > > > > > | --- | --- | --- | --- | --- | --- | --- | --- | --- |
> > > > > > | Horizons | 96 |  | 192 |  | 336 |  | 720 |  |
> > > > > > | Metrics | MSE | MAE | MSE | MAE | MSE | MAE | MSE | MAE |
> > > > > > | Constant | 0.167±0.002 | 0.223±0.001 | 0.182±0.002 | 0.238±0.001 | 0.197±0.002 | 0.249±0.002 | 0.203±0.002 | 0.262±0.001 |
> > > > > > | Xaiver | 0.167±0.002 | 0.222±0.001 | 0.184±0.002 | 0.241±0.002 | 0.194±0.002 | 0.252±0.001 | 0.191±0.002 | 0.248±0.002 |
> > > > > > | Kaiming | 0.168±0.002 | 0.224±0.001 | 0.181±0.002 | 0.234±0.002 | 0.188±0.003 | 0.244±0.002 | 0.193±0.001 | 0.248±0.002 |
> > > > > > | Randn | 0.167±0.002 | 0.222±0.001 | 0.182±0.003 | 0.237±0.001 | 0.188±0.002 | 0.245±0.003 | 0.195±0.002 | 0.251±0.002 |
> > > > > >
> > > > > > | Datasets | Traffic |  | Traffic |  | Traffic |  | Traffic |  |
> > > > > > | --- | --- | --- | --- | --- | --- | --- | --- | --- |
> > > > > > | Horizons | 96 |  | 192 |  | 336 |  | 720 |  |
> > > > > > | Metrics | MSE | MAE | MSE | MAE | MSE | MAE | MSE | MAE |
> > > > > > | Constant | 0.363±0.001 | 0.258±0.001 | 0.382±0.001 | 0.265±0.001 | 0.390±0.001 | 0.269±0.001 | 0.434±0.002 | 0.293±0.002 |
> > > > > > | Xaiver | 0.370±0.001 | 0.259±0.001 | 0.380±0.001 | 0.263±0.001 | 0.390±0.001 | 0.271±0.002 | 0.435±0.001 | 0.295±0.001 |
> > > > > > | Kaiming | 0.363±0.001 | 0.256±0.001 | 0.381±0.002 | 0.263±0.001 | 0.391±0.001 | 0.269±0.001 | 0.436±0.001 | 0.297±0.001 |
> > > > > > | Randn | 0.361±0.001 | 0.254±0.001 | 0.380±0.001 | 0.263±0.001 | 0.392±0.001 | 0.270±0.001 | 0.434±0.001 | 0.293±0.001 |

---

> ### Comment · Area_Chair_GEjv · 2025-08-05
>
> Dear Reviewer xNzv,
>
> Please help to go through the rebuttal and participate in discussions with authors. Thank you!
>
> Best regards,
> AC

---

> > ### Author Response · Authors · 2025-08-07
> > **Request to participate in the discussion**
> >
> > Dear Reviewer xNzv,
> >
> > Since the discussion phase ends soon, we sincerely request you to participate in the discussion. We hope our feedbacks could well handle your previous concerns, and if you have other questions we can also make further discussions!
> >
> > Best wishes!

---

> > > ### Comment · Reviewer_xNzv · 2025-08-09
> > > **Response**
> > >
> > > The responses have addressed my concerns and I will raise their scores to the level of Borderline accept.

---

> > > > ### Author Response · Authors · 2025-08-09
> > > > **Thanks for raising our scores!**
> > > >
> > > > Dear Reviewer xNzv, we are thrilled that our responses have effectively addressed your questions and comments. We would like to express our sincerest gratitude for taking the time to review our paper and provide us with such detailed feedback!

---

### Note · Authors · 2025-08-11

Dear Area Chair and reviewers:

Thanks again for your hard work and efforts! We are extremely grateful to all reviewers for their participation in the discussion, and we also appreciate the assistance from the Area Chair GEjv. During the discussion phase, **reviewer xNzv and Gwnu promise to improve the scores, and reviewer 7m7L and Hryh remain their positive attitude**. We are very excited that it seems all concerns have been effectively handled and all reviewers have a positive attitude towards our article, and we have included these valuable suggestions in our draft!

Authors of "Enhancing Time Series Forecasting through Selective Representation Spaces: A Patch Perspective"

Best Regards!

---

### Decision · Program_Chairs · 2025-09-17

**Decision:**

Accept (spotlight)

**Comment:**

This paper proposes the Selective Representation Space (SRS) module, a plug-and-play component that adaptively selects and reorders patches for time series forecasting, addressing the rigidity of conventional adjacent patching. The strengths lie in its novel and gradient-friendly design, consistent performance improvements across benchmarks, and the ability to enhance both a simple baseline and existing patch-based models. After rebuttal, the authors further clarified implementation details, provided more experiments and addressed concerns on efficiency. All the reviewers were satisfied with the rebuttal. Therefore, I would like to recommend accepting this paper.